# Overcoming resolution limits with quantum sensing

T. Gefen[1]*, A. Rotem[1] & A. Retzker [1]

The field of quantum sensing explores the use of quantum phenomena to measure a broad range of physical quantities, of both static and time-dependent types. While for static signals the main figure of merit is sensitivity, for time dependent signals it is spectral resolution, i.e. the ability to resolve two different frequencies. Here we study this problem, and develop new superresolution methods that rely on quantum features. We first formulate a general criterion for superresolution in quantum problems. Inspired by this, we show that quantum detectors can resolve two frequencies from incoherent segments of the signal, irrespective of their separation, in contrast to what is known about classical detection schemes. The main idea behind these methods is to overcome the vanishing distinguishability in resolution problems by nullifying the projection noise.

[1] Racah Institute of Physics, The Hebrew University of Jerusalem, Jerusalem 91904 Givat Ram, Israel. *email: tuvia.gefen@mail.huji.ac.il

Quantum metrology and quantum sensing[1,2] study parameter estimation limits in various physical systems by employing the fundamental laws of quantum physics. In particular this field seeks to optimize precision by utilizing quantum effects that have no classical analogs (such as entanglement and squeezing[3,4]).

A unique feature of quantum sensing is the ability to apply coherent control to the probe and vary the measurement basis. In particular this provides the ability to nullify the measurement projection noise. However, the contribution of this phenomenon to estimation problems has received scant attention.

In this paper, we highlight this feature and show that it is a critical resource primarily for resolution problems, that can improve precision by orders of magnitude. Resolution problems are ubiquitous and highly important in science[5–14], and roughly speaking are characterized by vanishing distinguishability; i.e, the sensitivity to the seperation between two close objects or frequencies vanishes as these get close enough. This effect usually results in divergent uncertainty, leading to a resolution limit. We show that it is possible to overcome the vanishing distinguishability by making the projection noise vanish as well, through a suitable control. These two effects can cancel each other out, leading to a finite uncertainty. We show that this is a general method to overcome resolution limits in quantum sensing.

Specifically, this method can be highly useful for analyzing complex spectrums with quantum sensors (such as quantum NMR problems[15–18]). An example for such a spectrum is illustrated in Fig. 1. While the two extreme frequencies can easily be estimated, the two central frequencies must be analyzed with a more sophisticated method, which eventually yields higher uncertainty. Here, we show that by using a quantum control, the spectrum can be shifted such that the projection noise vanishes. The vanishing projection noise implies a finite uncertainty irrespective of the frequency separation. In other words, the uncertainty does not diverge when the two frequencies merge. Furthermore, this method is extremely simple, unlike numerically demanding classical superresolution methods.

## Results

**Conditions for superresolution.** We first briefly review the pillars of quantum parameter estimation problems. A typical problem involves a quantum state $\rho(\theta)$, such that $\theta$ is to be estimated. The uncertainty in estimating $\theta$ is tightly lower bounded by $\frac{1}{\sqrt{I_\theta}}$, where $I_\theta$ is the Fisher information (FI) about $\theta$[19]. For a given choice of measurement of $\rho(\theta)$, $I_\theta$ is determined according to the probabilities $(p_j)$ in the following way: $I_\theta = \sum_j \frac{\left(\frac{dp_j}{d\theta}\right)^2}{p_j}$. The FI can be optimized over all possible measurements, leading to the quantum Fisher information (QFI)[20,21].

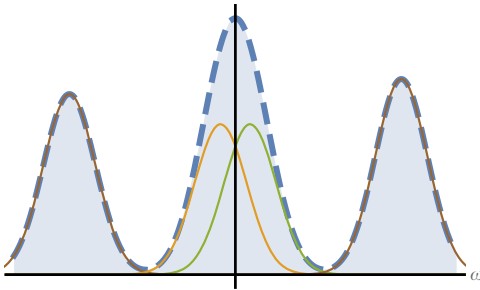

**Fig. 1** A spectrum analysis problem. While it is relatively easy to estimate the two side frequencies, the estimation of the two close frequencies is challenging and becomes infinitely difficult when the frequencies merge

Given a spectral decomposition $\rho = \sum_j p_j |\psi_j\rangle\langle\psi_j|$ the QFI about $\theta$ reads: $\mathscr{F} = \sum_{p_i+p_j\neq 0} \frac{2}{p_j+p_i} \left|\left(\frac{d\rho}{d\theta}\right)_{i,j}\right|^2$.

For a multivariable estimation of $\{\theta_k\}_k$, the error is quantified by the covariance matrix of the estimators. This covariance matrix is lower bounded by $\mathscr{F}^{-1}$, the inverse of the QFI matrix, where the QFI matrix is defined as $\mathscr{F}_{k,l} = 2\sum_{i,j} \frac{\left(\frac{\partial\rho}{\partial\theta_k}\right)_{i,j}\left(\frac{\partial\rho}{\partial\theta_l}\right)_{j,i}}{(p_i+p_j)}$.

We are now poised to formulate spectral resolution problems, which are the focus of this paper. In these problems we are given a signal (Hamiltonian) that oscillates with time. It consists of at most two frequencies, yet the exact number of frequencies (and their values) is unknown and need to be determined. To this end, a quantum probe interacts with the signal so that information about it becomes encoded on the probe and can be extracted by measurements. Once this information is extracted this problem boils down to a parameter estimation problem: the common strategy in these problems[6,13,22,23] is to assume that there are two frequencies and estimate them. If the estimation shows a significant overlap between the frequencies (significant with respect to the estimation error), it is concluded that the frequencies are not resolvable. However if the overlap is negligible, one can deduce that the signal consists of two frequencies (since the error probability is negligible). This implies that the figure of merit is $\Delta\omega_1, \Delta\omega_2$. The challenging regime is when $\omega_1 \to \omega_2$. Resolution becomes an issue when $\Delta\omega_1, \Delta\omega_2 \to \infty$ as $\omega_1 \to \omega_2$. A different, and somewhat more convenient, formulation uses $\omega_r = \frac{\omega_1-\omega_2}{2}, \omega_s = \frac{\omega_1+\omega_2}{2}$, so that the resolution condition is $\Delta\omega_r \ll \omega_r$ and the figure of merit is thus $\Delta\omega_r$. The key issue is thus the behavior of $\Delta\omega_r$ as $\omega_r \to 0$, if $\Delta\omega_r \to \infty$ then a fundamental resolution limit exists which is the case in relevant classical examples[6,13,22].

This limitation appears in various resolution problems (not only spectral resolution) and stems from a property of vanishing distinguishability. Let us define what vanishing distinguishability means. Given the quantum state of a probe (density matrix $\rho$), that depends on a set of parameters $\{\theta_i\}_i$, the state suffers from a vanishing distinguishability if the set $\left(\frac{\partial\rho}{\partial\theta_i}\right)_i$ is linear dependent. An equivalent way to define it: there exists a parameter $g$, that is a linear combination of $\{\theta_i\}_i$, such that $\frac{\partial\rho}{\partial g} = 0$. Indeed, in many resolution problems as the separation parameter $\omega_r$ (the difference between the frequencies or, in imaging, the sources) goes to 0, there exists a parameter $g$ such that $\frac{\partial\rho}{\partial g} = 0$. In this paper, we focus on the simplest (yet very common) case that only $\frac{\partial\rho}{\partial\omega_r} = 0$ as $\omega_r = 0$. In this case $\Delta\omega_r \to \infty$ if and only if the FI about $\omega_r$ (denoted as $I_r$) vanishes, which implies that $I_r$ is our figure of merit.

As an example, consider a signal that acts on a qubit and is given by the following Hamiltonian:

$$H = [A_1\cos(\omega_1 t) + B_1\sin(\omega_1 t) + A_2\cos(\omega_2 t) + B_2\sin(\omega_2 t)]\sigma_z.$$

$$(1)$$

It is simple to see that this limitation appears whenever the Hamiltonian posses a symmetry for exchange of $\omega_1 \leftrightarrow \omega_2$ (i.e. identical amplitudes). This symmetry implies a symmetry of $\omega_r \leftrightarrow -\omega_r$, from which it follows that the state obtained after evolution time $t$ has the same symmetry, $|\psi_t(\omega_r)\rangle = |\psi_t(-\omega_r)\rangle$, and thus $\frac{\partial|\psi_t\rangle}{\partial\omega_r} = 0$ for $\omega_r = 0$. Given the expression of the QFI[21],

we obtain:

$$I_r \leq 4\left[\left\langle\frac{\partial\psi}{\partial\omega_r}\Big|\frac{\partial\psi}{\partial\omega_r}\right\rangle - \left|\left\langle\frac{\partial\psi}{\partial\omega_r}\Big|\psi\right\rangle\right|^2\right] \to 0, \qquad (2)$$

hence resolution is limited. Note that applying further control on the probe cannot eliminate this symmetry, and thus cannot remove this resolution limit.

It can be shown that this limitation appears for any quadratures: there exists a parameter $g$ such that $\frac{\partial H}{\partial g} = 0$ for every $t$, which implies $\frac{\partial|\psi_t\rangle}{\partial g} = 0$ for every measurement (more details in Supplementary Note 4).

So vanishing distinguishability is quite a common property and appears in different resolution problems, but does it always impose a limitation?

Eq. (2) shows that whenever the quantum state of the probe ($\rho$) is pure, resolution is limited, however for some mixed states this property does not limit the resolution, these are the states that give rise to superresolution: $\frac{d\rho}{d\omega_r} \to 0$ yet $I_r(\omega_r \to 0) > 0$. A special case of this phenomenon was found and analyzed recently in the context of optical imaging[10–12,24] (see Supplementary Note 3). Can such states be obtained in quantum spectroscopy and various other problems? In order to understand this, it would be highly desirable to characterize these states and set a sharp condition for superresolution.

Let us show that these states can be simply characterized:

**Claim** *Given $\rho(\omega_r)$ such that $\frac{d\rho}{d\omega_r} = 0$ (as $\omega_r \to 0$), then $I_r(\omega_r \to 0) > 0$ if and only if at least one of the eigenvalues of $\rho$ goes as $\omega_r^k$, where $1 < k \leq 2$ or equivalently $\frac{d\sqrt{\rho}}{d\omega_r} \neq 0$. The optimal measurement basis converges to an eigenbasis of $\rho$ as $\omega_r \to 0$.*

We briefly illustrate a proof: Given a spectral decomposition $\rho = \sum_j p_j|j\rangle\langle j|$, then:

$$\frac{d\rho}{d\omega_r} = \sum_j \frac{dp_j}{d\omega_r}|j\rangle\langle j| + i\sum_{j,k}(p_j - p_k)h_{k,j}|k\rangle\langle j|, \qquad (3)$$

where $h$ is a Hermitian operator and $h_{k,j}$ denote its matrix elements in the eigenbasis of $\rho$. Since $\frac{d\rho}{d\omega_r} \to 0$, then for every $j,k$: $\frac{dp_j}{d\omega_r} \to 0$, $(p_j - p_k)h_{k,j} \to 0$. With this notation, the QFI ($\mathscr{F}$) reads (see ref. [21]):

$$\mathscr{F} = \sum_j \frac{\left(\frac{dp_j}{d\omega_r}\right)^2}{p_j} + 2\sum_{j,k}\frac{(p_j - p_k)^2}{p_j + p_k}|h_{kj}|^2. \qquad (4)$$

The fact that $(p_j - p_k)h_{k,j} \to 0$ implies that $\frac{(p_j-p_k)^2}{p_j+p_k}|h_{kj}|^2 \to 0$, however $\frac{dp_j}{d\omega_r} \to 0$ does not imply that $\frac{\left(\frac{dp_j}{d\omega_r}\right)^2}{p_j}$ vanishes. It can be seen that given that $\frac{dp_j}{d\omega_r} \to 0$, $\frac{\left(\frac{dp_j}{d\omega_r}\right)^2}{p_j} > 0$ if and only if there exists $p_j \sim \omega_r^k$ for $1 < k \leq 2$. We then observe that for $\omega_r \to 0$, $\mathscr{F}(\rho) \to \sum_j \frac{\left(\frac{dp_j}{d\omega_r}\right)^2}{p_j}$, which implies that the optimal measurement basis converges to any eigenbasis of $\rho$.

This condition can be shown to be equivalent to $\frac{d\sqrt{\rho}}{d\omega_r} \neq 0$ (see Supplementary Note 1). It is quite intuitive that one has to demand $\frac{d\sqrt{\rho}}{d\omega_r} \neq 0$, since the QFI equals the minimization of all the QFI's of the purifications. Since purifications go as $\sqrt{\rho}$, $\frac{d\sqrt{\rho}}{d\omega_r} = 0$ would imply a vanishing derivative of every purification and thus a vanishing QFI.

This criterion shows that the only way to overcome a vanishing distinguishability is by nullifying the projection noise of one of the outcomes.

This condition is a special case of a more general (multivariate) criterion. In the multivariate version $\left(\frac{\partial\rho}{\partial\theta_i}\right)_{i=1}^n$ are linearly dependent (with dimension $k < n$) and the relevant question is whether the QFI matrix can be regular. Note that we can choose $(\theta_i)_{i=1}^n$ such that $\left(\frac{\partial\rho}{\partial\theta_i}\right)_{i=1}^k$ are linearly independent and $\frac{\partial\rho}{\partial\theta_{k+1}} = \ldots = \frac{\partial\rho}{\partial\theta_n} = 0$ ($\theta_{k+1}, \ldots, \theta_n$ are the problematic parameters). Then the QFI is regular if and only if the classical FI matrix (i.e. the FI matrix obtained when measuring in the eigenbasis of $\rho$) about the problematic parameters ($\theta_{k+1}, \ldots, \theta_n$) is regular. Namely it depends only on the classical FI about these parameters, and thus the optimal measurement basis to estimate these parameters is the eigenbasis of $\rho$. The proof of this condition is quite similar to that of the single variable case, and is given in Supplementary Note 2.

Before we move on to applications in quantum sensing, a few remarks are in order: An accurate formulation of the super-resolution condition is $\frac{d\rho}{d\omega_r} \to 0$ and $I_r(\omega_r \to 0) > 0$, namely the limit needs to be positive. That is because we are interested in the behavior of the FI for a very small difference, rather than a vanishing difference. We mention this point since the FI at $\omega_r = 0$ can be discontinuous or meaningless (Cramer–Rao bound may be violated), as one of the eigenvalues vanishes[25,26]. Given a vanishing eigenvalue, the variance of maximum likelihood estimation will vanish (which corresponds to an infinite FI) and thus may not coincide with the limit.

We also remark that in all cases examined in this paper (as well as in the imaging case) the eigenvalue goes as $\omega_r^2$. Any different power, $1 < k < 2$, would in fact lead to a better performance: a divergent FI.

**Application: spectral resolution without coherence**. Consider now again the problem of spectral resolution, with the signal defined in Eq. (1), and such that it suffers from shot-to-shot noise: in each measurement the frequencies are the same but the quadratures are random, i.e. $A_i, B_i$ have a certain distribution. Specifically here we assume $A_i, B_i \sim N(0, \sigma)$, and other noise models are addressed in Supplementary Note 11. This scenario is illustrated in Fig. 2, and is relevant for different applications, such as communication protocols, spectrum analyzers and nano NMR[15,16,27–35], in particular when the time required to perform projective measurement is longer than the coherence time of the signal (this is the case with NV centers, due to the large number of iterations needed, and with trapped ions, where the re-cooling process might be longer than the coherence time of the qubit).

It is quite clear that the fluctuations of the quadratures remove the purity of the probe, which can give rise to superresolution states. Let us examine this.

Consider a standard Ramsey experiment, in which the probe is initialized in $\sigma_x - \sigma_y$ plane, then rotated due to the signal and eventually measured in the initialization basis. Due to the fluctuations of the Hamiltonian, an averaging should be performed. Therefore the state of the probe is given by a density matrix:

$$\rho = \int p(A_i)p(B_i)|\psi_{A_i,B_i}\rangle\langle\psi_{A_i,B_i}|\, dA_i dB_i, \qquad (5)$$

where $|\psi_{A_i,B_i}\rangle$ is the state given a single realization of $A_i, B_i$. Note

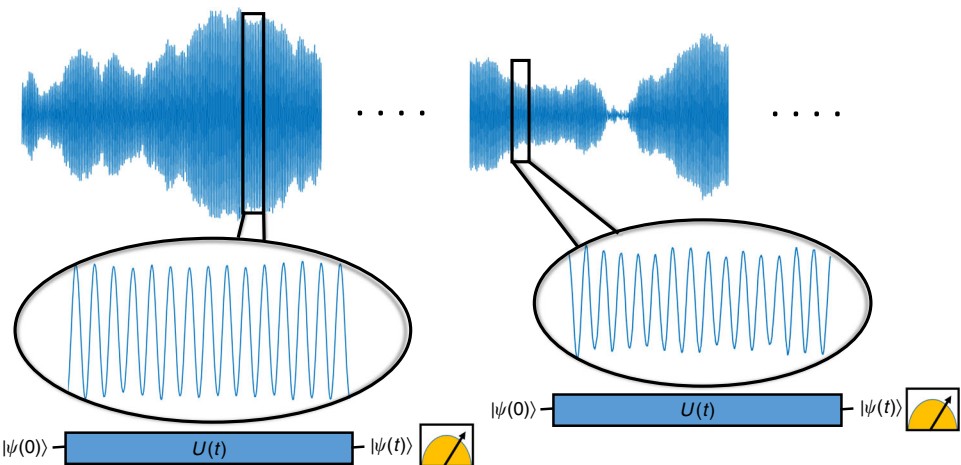

**Fig. 2** The problem of resolution without coherence. The quadratures of the signal in different measurements are random. The question we address is whether resolution is limited by the length of individual measurements

that since the fluctuations are identical:

$$\rho(\omega_r) = \rho(-\omega_r) \Rightarrow \frac{d\rho}{d\omega_r} = 0 \; (\omega_r = 0). \tag{6}$$

Once again, control on the probe does not change this symmetry, hence superresolution can be achieved only if the condition presented above is satisfied: projection noise has to be nullified. It is therefore desirable to find a measurement scheme that nullifies the projection noise. It is simple to see that this can be obtained if $\phi_{A_i,B_i} = 0 (\forall A_i, B_i)$, where $\phi_{A_i,B_i}$ is the phase accumulated by the sensor (defined as half the rotation angle in the Bloch sphere) per realization, since this implies a vanishing transition probability.

Our claim is therefore: Given the above noise model, there exist measurement schemes that satisfy the superresolution condition and thus achieve $I_r > 0$. To see that such methods exist, observe that the phase accumulated by the sensor given the Hamiltonian in Eq. (1) (when no control is applied) reads:

$$\phi_{A_i,B_i} = \sum_i \frac{A_i}{\omega_i} \sin(\omega_i t) + \frac{B_i}{\omega_i}(1 - \cos(\omega_i t)). \tag{7}$$

Note that given this time evolution the density matrix of the sensor is diagonal in the initialization basis with eigenvalues $p$, $1 - p$, where $p$ is the average transition probability: $p = \langle \sin(\phi_{A_i,B_i})^2 \rangle_{A_i,B_i}$. Hence the superresolution condition boils down to $p \sim \omega_r^2$. This indeed can be satisfied by simply tuning $t$ such that $\omega_s t = 2\pi n$, where $n$ is a non-zero integer. With this tuning $\phi_{A_i,B_i} = 0$ (for $\omega_r = 0$), and more specifically :

$$\phi_{A_i,B_i} \approx \frac{(A_1 - A_2)}{\omega_s} \omega_r t \rightarrow p \approx \frac{2\sigma^2}{\omega_s^2} \omega_r^2 t^2, \tag{8}$$

Hence the superresolution condition is satisfied and the FI reads:

$$I_r = \frac{8\sigma^2 t^2}{\omega_s^2}. \tag{9}$$

So nullifying the projection noise indeed cancels the vanishing derivative and a finite $I_r$ is achieved.

The obtained FI can be still quite poor and far from optimal. Note that it goes as $1/n^2$, where $n$ is the number of periods completed during the measurement. If $n$ is large, then this factor of $\frac{1}{n^2}$ can be significant. A much better FI can be achieved by applying a suitable control: $\pi$−pulses which effectively change the frequency of oscillations, and reduce $n$ to $1$[36–39]:

Given an original Hamiltonian of $H = [A \sin(\omega t) + B \cos(\omega t)]\sigma_z$, applying $\pi$−pulses in a frequency of $\omega + \delta$ (namely a $\pi$−pulse is applied every $\frac{\pi}{\omega + \delta}$, $\delta$ is referred to as detuning) on the probe yields the following effective Hamiltonian (see the "Methods" section for a derivation):

$$H_{eff} = \tan\left(\frac{\pi}{2\left(1 + \frac{\delta}{\omega}\right)}\right)\left(\frac{\delta}{\omega}\right)[A \sin(\delta t) + B \cos(\delta t)]\sigma_z. \tag{10}$$

Hence the $\pi$−pulses effectively change the frequency of the Hamiltonian from $\omega$ to $\delta$ (with a prefactor of $\tan\left(\frac{\pi}{2\left(1 + \frac{\delta}{\omega}\right)}\right)\left(\frac{\delta}{\omega}\right)$ added to the amplitude). Since we aim to reduce the frequency of oscillations, we focus on the limit of $\delta \ll \omega$, in which $\tan\left(\frac{\pi}{2\left(1 + \frac{\delta}{\omega}\right)}\right)\left(\frac{\delta}{\omega}\right) \approx \frac{2}{\pi}$, and thus:

$$H_{eff} \approx \frac{2}{\pi}[A \sin(\delta t) + B \cos(\delta t)]\sigma_z. \tag{11}$$

When dealing with a signal that consists of two frequencies $(\omega_1, \omega_2)$, the effective Hamiltonian becomes:

$$H_{eff} \approx \sum_i \frac{2}{\pi}[A_i \sin(\delta_i t) + B_i \cos(\delta_i t)]\sigma_z. \tag{12}$$

Hence due to the control the central frequency is shifted to $\delta_s = \frac{\delta_1 + \delta_2}{2}$, and the relative frequency simply changes sign: $\delta_r = -\omega_r$. The condition of vanishing $p$ becomes: $\delta_s t = \pm 2\pi n$, such that the optimal strategy is setting $\delta_s t = \pm 2\pi$. Therefore with these (optimal) values of $\delta_s$ the FI reads:

$$I_r \approx \left(\frac{2}{\pi}\right)^2 \frac{8\sigma^2 t^2}{\delta_s^2} = \frac{8\sigma^2 t^4}{\pi^4}. \tag{13}$$

Observe that the scaling of $I_r$ is optimal (goes as $\sigma^2 t^4$)[37,39,40,40–43]; however, it is unknown whether this is the best achievable FI (see extended discussion in Supplementary Note 6). The probabilities and the FI for different detunings are presented in Fig. 3. Note that clear resonance peaks of the FI are observed for $\delta_s t = \pm 2\pi n$, any other values of detuning lead to a vanishing FI.

We tested this method numerically by generating data of two frequency signal (with the corresponding noise model) and performing a Maximum-likelihood estimation (MLE) to find $\omega_r$. Some of the results are shown in Fig. 4. It can be seen that by choosing a detuning such that $\delta_s t = 2\pi$, $\omega_r$ can be estimated efficiently and the frequencies are resolved. As shown in Fig. 4, the standard deviation matches the theoretical expectation: $\Delta\omega_r = \frac{1}{\sqrt{I_r N}}$. By utilizing this control method the number of

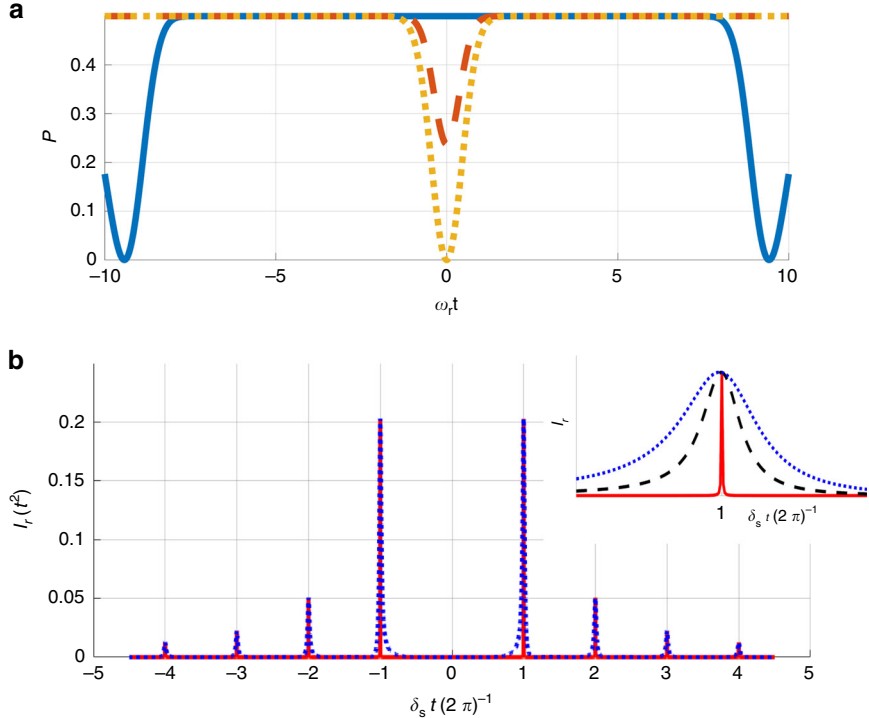

**Fig. 3** Probability and Fisher information analysis. **a** Average transition probability ($p$) as a function of $\omega_r t$ for different values of $\delta_s$: $\delta_s t = \pi$ (blue, solid line), $\delta_s t = 1.8\pi$ (orange, dashed line) and $\delta_s t = 2\pi$ (dotted, yellow line). For every $\delta_s$, $\frac{dp}{d\omega_r} = 0$ for $\omega_r = 0$. Hence a finite $I_r$ can be achieved only if $p = 0$. This requirement is fulfilled when $\delta_s t = 2\pi n$. **b** FI about $\omega_r$ ($I_r$) as a function of $\delta_s t$. Clear peaks can be observed whenever $\delta_s t = 2\pi n$. The width of the peaks is illustrated in the inset: for $\omega_r = 0$, the width vanishes; however finite $\omega_r$ leads to a finite width (given $\omega_r t \ll 1$ this width goes as $\omega_r$, see section "Limitations and imperfections"). For this illustration: $\omega_r t = 0.001$ (red, solid line), $\omega_r t = 0.05$ (black, dashed line), $\omega_r t = 0.1$ (blue, dotted line)

measurements ($N$) needed to achieve resolution is $N \gg p^{-1} = \frac{\pi^4}{2\sigma^2 \omega_r^2 t^4}$. Taking for example values which are well beyond the resolution limit, such as $\omega_r t = 0.01$, $\sigma t = 1$, resolution is achieved for $N \gg 5 \times 10^5$. If the chosen detuning does not satisfy one of these conditions ($\delta_s t = 2\pi n$) we expect to observe a divergence in the variance. We used MLE for this case as well. Note that the fact that the FI vanishes does not mean that no information about $\omega_r$ is obtained, information is in fact obtained from the second derivative. The estimator becomes biased and the standard deviation reads: $\Delta\omega_r = \frac{(p(1-p))^{0.25}}{\sqrt{\frac{\partial^2 p}{\partial \omega_r^2}} N^{0.25}}$, (see Fig. 4c). The fact that the standard deviation is proportional to $N^{-0.25}$ (as opposed to the standard scaling of $N^{-0.5}$) is a manifestation of the divergence. The resolution condition in this case is thus: $N \gg \frac{p(1-p)}{\left(\frac{\partial^2 p}{\partial \omega_r^2}\right)^2 \omega_r^4}$. Considering the same example as previously ($\omega_r t = 0.01$, $\sigma t = 1$) but with off-resonance detuning ($\delta_s = 1.8\pi$), the number of measurements required for resolution is $N \gg 10^8$; hence a difference of almost three orders of magnitude.

This method can be understood in the following simple and intuitive way: If there is only a single frequency and $\delta_s = \frac{2\pi}{t}$ then $p_a = 0$, hence no transitions should occur, whereas a finite (small) $\omega_r$ should lead to a small transition probability ($p$), such that transitions will be observed after $\frac{1}{p} = \frac{\pi^4}{2\sigma^2 \omega_r^2 t^4}$ measurements.

**Limitations and imperfections**. The method, as analyzed so far, assumes knowledge of all the other parameters ($\sigma$ and $\omega_s$), coherence of the signal and the probe during the measurement period, and measurements with unit fidelity. In this section, we

analyze each one of these assumptions. The first one to be analyzed is the main caveat of the method: the requirement of coherence during the measurement period.

This method relies on the ability to nullify projection noise, in particular on the fact that for $\omega_r = 0$ the state can become pure. However, this is achieved only for a signal which is perfectly coherent during each measurement. Fluctuations of the signal during the measurement period inflict a limitation, as in this case it is not possible to nullify the projection noise. Heuristically due to these fluctuations the transition probability includes an additional noise term (denoted as $\epsilon$), such that it reads (for $\omega_s t = 2\pi$):

$$p = \frac{\sigma^2 t^2}{2\pi^2} \omega_r^2 t^2 + \epsilon. \qquad (14)$$

This new term imposes a limitation: It is now impossible to nullify $p$, which implies that $I_r \rightarrow 0$ as $\omega_r \rightarrow 0$. The FI (for $\omega_s t = 2\pi$, $\omega_r t \ll 1$) now reads:

$$I_r \approx \frac{(\sigma t)^2 (\omega_r t)^2}{\pi^4 \left(\epsilon + \frac{\sigma^2 t^2}{2\pi^2} \omega_r^2 t^2\right)}, \qquad (15)$$

this behavior is illustrated in Fig. 5, and it can be observed that resolution can be achieved only for $\omega_r t > \frac{\sqrt{\epsilon}}{\sigma t}$.

More specifically, assuming a realistic noise model: the quadratures undergo Ornstein–Uhlenbeck (OU) noise process (with variance $\sigma_n^2$ and damping rate $\gamma$), the noise term reads (in leading order of $(\gamma t \ll 1)$, see Supplementary Note 8) $\epsilon = \frac{\sigma_n^2 t^3}{\pi^2}$. When comparing $\epsilon$ to the original transition probability: $\sigma^2 \frac{t^2}{2\pi^2} \omega_r^2 t^2 = \frac{\sigma_n^2 t^2}{4\pi^2 \gamma} \omega_r^2 t^2$, we get the Fourier limit: $\frac{\omega_r}{\gamma} > 1$. We remark that whether one can remove this limitation is an open question.

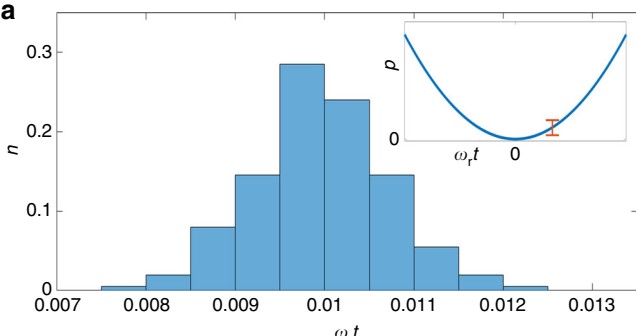

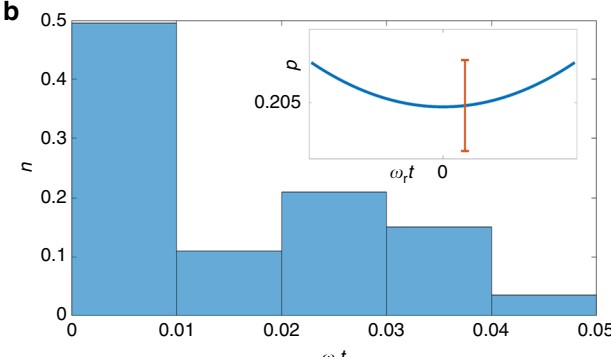

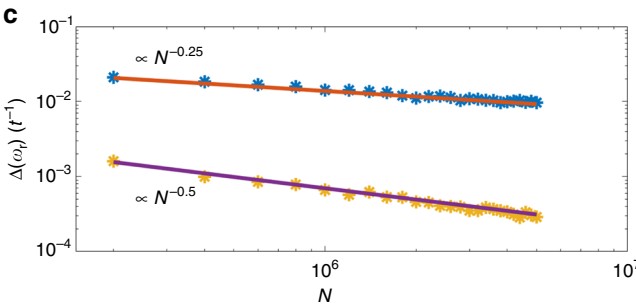

**Fig. 4** Estimation analysis. Maximum-likelihood estimation of $\omega_r$ (beyond the resolution limit; i.e. $\omega_r t \ll 1$) with different control methods. **a** and **b** Histogram of the estimated $\omega_r$ for the optimal control method: $\delta_s t = 2\pi$, compared to the histogram obtained slightly off the resonance: $\delta_s t = 1.8\pi$. When resonance is achieved, the two frequencies are clearly resolved ($\Delta\omega_r < \frac{1}{10}\omega_r$), while off the resonance they are not resolvable ($\Delta\omega_r > \omega_r$). Note that off resonance, the standard deviation is too large; hence the probability cannot be distinguished from $p(\omega_r = 0)$ (see insets). For both plots $N = 10^6$, $\sigma t = 5$, $\omega_r t = 0.01$. **c** The root mean square error (RMSE) as a function of $N$ for both control methods. For $\delta_s t = 2\pi$ the RMSE goes as $(N I_r)^{-0.5}$ as expected. Off the resonance ($\delta_s t = 1.8\pi$) the FI vanishes and the RMSE goes as $N^{-0.25}$ (the estimation is biased)

Therefore this method is relevant mainly for experimental scenarios with noise that is effectively shot to shot: small enough fluctuations during each measurement but no correlations between consecutive measurements. This is the case in many experimental settings, where the time separation between measurements is longer than the phase acquisition period due to long readout and preparation stages.

Quite similarly, dephasing of the probe also imposes a limitation. Taking into account a dephasing rate $\kappa$, the transition probability reads: $p = 0.5\left(1 - \exp\left(-\frac{(\sigma t)^2(\omega_r t)^2}{\pi^2} - 2\kappa t\right)\right)$. Hence resolution can be achieved only if $\frac{\omega_r \sigma}{(\kappa^2)} \gg 1$. Note that in order to retrieve the noiseless FI it is not enough to require $\kappa t \ll 1$, as

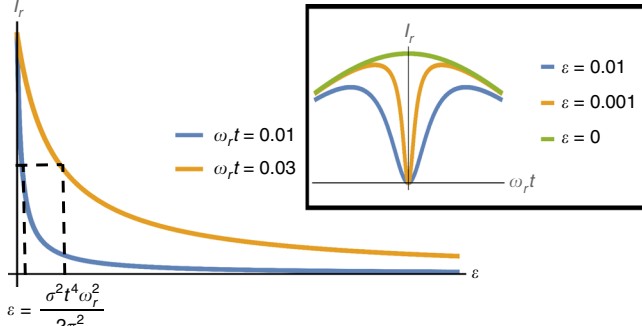

**Fig. 5** The effect of noise. $I_r$ as a function of $\epsilon$ (a general noise term in the measurement, see Eq. (14)) for different values of $\omega_r$. $I_r$ drops to half the maximal value for $\epsilon = \frac{(\sigma t)^2(\omega_r t)^2}{2\pi^2}$, which means that the maximal $\epsilon$ for which resolution can be achieved goes as $(\sigma t)^2(\omega_r t)^2$. In the inset: $I_r$ as a function of $\omega_r$ for different $\epsilon$

there is also minimal time $t > \frac{\kappa^{1/3}}{\sigma^{2/3}\omega_r^{2/3}}$ ($\kappa t$ should be smaller than $(\sigma t)^2(\omega_r t)^2$). A detailed analysis of this limitation can be found in Supplementary Note 9.

We next address the consequences of imperfect measurements. The effect of imperfect measurements is similar to that of incoherence, therefore the measurement infidelity sets a resolution limit. We consider a model in which there are two different outcomes and there is a finite probability to get each outcome from both states (as is the case for the NV center[44]). Namely the probability of detecting an outcome that corresponds to the bright state is: $p = (1 - \epsilon')p_b + \epsilon' p_d$, where $p_b$ ($p_d$) denotes the probability of the bright (dark) state and then $\epsilon'$ is the probability of wrong detection. Given this error probability we can observe that $\frac{dp}{d\omega_r} = 0$ (when $\omega_r = 0$) but it is impossible to nullify $p$. This implies $I_r \rightarrow 0$ as $\omega_r \rightarrow 0$. Therefore taking $\epsilon' \ll 1$ (and $\omega_r t \ll 1$, $\omega_s t = 2\pi$) we get the same expression as in Eq. (14) (with a noise term of $\epsilon'$): $p \approx \frac{\sigma^2 T^2}{2\pi^2}\omega_r^2 T^2 + \epsilon'$. Hence the resolution limit is given by: $\omega_r T > \frac{\sqrt{\epsilon'}}{\sigma T}$ (see Fig. 5).

Let us now address the multivariable estimation protocol. In any realistic scenario $\sigma$ and $\omega_s$ are unknown. Since the estimation protocol of $\omega_r$ depends on knowledge of $\omega_s$ a preliminary estimation of $\omega_s$ must be performed (quite analogously to the preliminary estimation of the centroid in quantum resolution methods for optical imaging[45,46]). This can be done using the traditional method[17]: Applying $\pi$-pulses in different frequencies and fitting the transition probability as a function of the pulses frequency (see Supplementary Note 7). This should provide a good estimation of $\sigma$, $\omega_s$, but not a good enough estimation of $\omega_r$ (unless by chance we hit close enough to a resonance frequency). Once a good enough estimation of $\omega_s$ is obtained we can apply the required control ($\delta_s t = 2\pi$). To understand what is a good enough estimation of $\omega_s$ observe that for small enough $\omega_r t$, ($\delta_s t - 2\pi$): $I_r \approx \frac{8\sigma^2 t^4}{\pi^4}\frac{\omega_r^2}{\omega_r^2 + (\delta_s - 2\pi/t)^2}$, hence the width of the resonance peak (in $\delta_s t$) goes as $\omega_r$. Therefore once $\Delta\omega_s$ is comparable to $\omega_r$ this method works despite the small detuning.

Observe that now a multivariate estimation should be performed, which means that at least three different measurements are needed; each measurement in a detuning that is optimal for a different parameter. Numerical results and further analysis are presented in Supplementary Note 7.

**Additional applications: quantum resolution methods for sampling.** Superresolution with quantum Fourier transform (QFT): Consider the signal (Hamiltonian) in Eq. (1), if the

coherence time of the signal is relatively long, its spectrum can be found by sampling it (where sampling means Ramsey measurements of the probe in different times). Several recent experiments implemented this scheme[37,38,47]. The straightforward (and natural) way to analyze this data is by fitting the power spectrum of the measurement outcomes, however it was shown in ref. [23] that this method suffers from a resolution limit. The reason is again that the (average) power spectrum is symmetric with respect to $\omega_r$, yet the measurement noise does not vanish. We point out here that this limit can be eliminated if instead of classical Fourier transform, one uses a QFT. In more detail: a phase of $\phi_j \approx \tau \sum_i (A_i \cos(\omega_i t_j) + B_i \sin(\omega_i t_j))$ is accumulated by the probe in each measurement (where $\tau$ is the length of each measurement). The idea is that instead of measuring the probe after each phase acquisition, we can map the phases to memory qubits to form the state: $|\psi\rangle = \frac{1}{\sqrt{N}} \sum_{j=1}^{N} e^{i\phi_j} |j\rangle$, and then measure in the Fourier basis. With the appropriate choice of total sampling time ($T = m \frac{2\pi}{\omega_s}$, for an integer $m$) and measurement length ($\tau = \frac{2\pi}{n\omega_s}$, for an integer $n$), the only states in the Fourier basis that can be measured are harmonics of $\omega_s$ (for $\omega_r = 0$), hence vanishing projection noise of all the other outcomes. The probability to measure the other frequencies, for $\omega_r T \ll 1$, is $\approx \frac{1}{6} \omega_r^2 T^2 (\sigma\tau)^2$ (see supplementary note 10), therefore $I_r = \frac{2}{3}(\sigma\tau)^2 T^2$ for $\omega_r = 0$. We thus get a non-vanishing FI, and it can be shown that for any model in which the phases are uniformly distributed, the optimal measurement basis is indeed the Fourier basis.

A different method for the same problem is superresolution with correlation spectroscopy: The Hamiltonian is the same, but now perform two measurements and correlate between them using a single memory qubit, namely the state of the memory qubit after the two phase accumulation periods is $\frac{1}{\sqrt{2}}\left(|0\rangle + e^{i(\phi_1 - \phi_2)}|1\rangle\right)$, where $\phi_j$ is the phase accumulated in the $j$th period. It is simple to see that by choosing the period between measurements to be $T = \frac{2\pi}{\omega_s}n$, and measuring in the initialization basis the transition probability is $p \approx \langle(B_1 - B_2)^2\rangle \tau^2 \omega_r^2 T^2$. Therefore a non-vanishing FI is achieved: $I_r = 4\langle(B_1 - B_2)^2\rangle \tau^2 T^2 = 8(\sigma\tau)^2 T^2$.

## Discussion

We presented methods that are capable of resolving frequencies beyond the resolution limits ($\omega_r t \ll 1$) in quantum spectroscopy. Those methods are special cases of a general superresolution criterion: one can overcome the vanishing derivative by making the projection noise vanish at the same rate. The main method that was analyzed (resolution without coherence) is applicable with state of the art experimental capabilities and does not require involved numerical analysis.

It would be interesting to inquire whether similar ideas are relevant to other resolution problems, such as resolving the locations and the frequencies of single neighboring spins.

The methods presented above are not perfect, they are limited by the noise of the signal and the dephasing of the probe, whether one can overcome these limitations is an open question.

## Methods

**Derivation of density matrix and probabilities.** Given a noise model on the amplitudes, the quantum state of the probe is described by the following density matrix:

$$\rho = \int |\psi\rangle\langle\psi| p(\mathbf{A}, \mathbf{B}) d\mathbf{A} \; d\mathbf{B}. \tag{16}$$

Since the time evolution (with and without control) is described by the operator:

$U = \cos(\phi)\mathbb{1} - i\sin(\phi)\sigma_z$, $\rho$ reads:

$$\rho = \int \left[ \cos(\phi)^2 \rho_0 + \sin(\phi)^2 \sigma_z \rho_0 \sigma_z - \frac{i}{2}\sin(2\phi)[\sigma_z, \rho_0] \right] \cdot p(\mathbf{A}, \mathbf{B}) d\mathbf{A} \; d\mathbf{B}, \tag{17}$$

where $\rho_0$ is the initial state. With the relevant noise model ($A_i, B_i \sim N(0, \sigma)$) it can be seen that the terms going as $\sin(2\phi)$ vanish, leading to:

$$\rho = (1 - p)\rho_0 + p\sigma_z\rho_0\sigma_z, \tag{18}$$

where $p$ is the (averaged) transition probability: $\int \sin(\phi)^2 p(\mathbf{A}, \mathbf{B}) d\mathbf{A} \; d\mathbf{B}$. Taking $\phi = \sum_i A_i \frac{\sin(\delta_i t)}{\delta_i} + B_i \frac{1 - \cos(\delta_i t)}{\delta_i}$, a simple calculation yields:

$$p = 0.5\left(1 - \exp\left(-8\sum_i \frac{\sigma^2}{\delta_i^2}\sin^2\left(\frac{\delta_i t}{2}\right)\right)\right). \tag{19}$$

Note that this expression coincides with Eq. (8) for $\delta_s t = 2\pi$, $\omega_r t \ll 1$. The optimal initial state would be $\rho_0 = |\uparrow_x\rangle\langle\uparrow_x|$ (or any other pure state in the $X - Y$ plane), leading to $\rho = (1 - p)|\uparrow_x\rangle\langle\uparrow_x| + p|\downarrow_x\rangle\langle\downarrow_x|$. The QFI (about $\omega_r$) of $\rho$ is thus: $\frac{\left(\frac{dp}{d\omega_r}\right)^2}{p(1-p)}$, which is the expression of $I_r$ mentioned in the main text.

**Effective Hamiltonian derivation.** In this section, we derive the effective Hamiltonian that appears in the main text. Given a Hamiltonian: $H = [A\sin(\omega t) + B\cos(\omega t)]\sigma_z$, and $\pi$-pulses that are applied every $\tau$, the Hamiltonian in the interaction picture of these pulses is

$$H = [A\sin(\omega t) + B\cos(\omega t)]h(t)\sigma_z, \tag{20}$$

where $h(t)$ is the square wave function. Note that the phase accumulated by the sensor (denoted as $\phi$, and defined as half the rotation angle in Bloch sphere) in $t = n\tau$ is

$$\phi = A\,\text{Im}(\Phi) + B\,\text{Re}(\Phi) \text{ where } \Phi = \sum_{n=0}^{N-1} \int_{n\tau}^{(n+1)\tau} e^{i\omega t}(-1)^n dt \tag{21}$$

where Re (Im) denotes the real (imaginary) part. Therefore in order to find $\phi$ we need to calculate $\Phi$:

$$\Phi = \sum_{n=0}^{N-1} e^{i\omega n\tau}(-1)^n \frac{e^{i\omega\tau} - 1}{i\omega} = \sum_{n=0}^{N-1} e^{in(\omega\tau + \pi)} \frac{e^{i\omega\tau} - 1}{i\omega}. \tag{22}$$

The calculation then proceeds as follows:

$$\Phi = \frac{1 - e^{iN(\omega\tau + \pi)}}{1 + e^{i\omega\tau}} \frac{e^{i\omega\tau} - 1}{i\omega}$$
$$= -2ie^{\frac{iN}{2}(\omega\tau + \pi)}\sin\left(N\frac{\omega\tau}{2} + N\frac{\pi}{2}\right)\sin\left(\omega\frac{\tau}{2}\right)\frac{1}{\cos\left(\frac{\omega\tau}{2}\right)\omega}. \tag{23}$$

Hence

$$\text{Re}(\Phi) = (1 - \cos(\omega t + N\pi))\frac{\sin\left(\omega\frac{\tau}{2}\right)}{\cos\left(\omega\frac{\tau}{2}\right)\omega},$$
$$\text{Im}(\Phi) = -\sin(\omega t + N\pi)\frac{\sin\left(\omega\frac{\tau}{2}\right)}{\omega\cos\left(\omega\frac{\tau}{2}\right)}. \tag{24}$$

Note that: $\omega t = \omega N \frac{\pi}{\omega + \delta} = N\pi - \delta t$, therefore Eq. (24) is simplified to:

$$\text{Re}(\Phi) = (1 - \cos(\delta t))\frac{\tan\left(\omega\frac{\tau}{2}\right)}{\omega},$$
$$\text{Im}(\Phi) = \sin(\delta t)\frac{\tan\left(\omega\frac{\tau}{2}\right)}{\omega}. \tag{25}$$

The accumulated phase, $\phi$, thus reads:

$$\phi = A\sin(\delta t)\frac{\tan\left(\omega\frac{\tau}{2}\right)}{\omega} + B(1 - \cos(\delta t))\frac{\tan\left(\omega\frac{\tau}{2}\right)}{\omega}. \tag{26}$$

Observe that this exact phase is obtained by the following effective Hamiltonian (note that no approximation is used here):

$$H_{\text{eff}} = \tan\left(\omega\frac{\tau}{2}\right)\left(\frac{\delta}{\omega}\right)[A\cos(\delta t) + B\sin(\delta t)]\sigma_z, \tag{27}$$

hence we can use this effective Hamiltonian to describe the dynamics. This effective Hamiltonian is somewhat similar to the original Hamiltonian in that the frequency is shifted from $\omega$ to $\delta$, and the amplitude acquires a prefactor of $\tan\left(\omega\frac{\tau}{2}\right)\left(\frac{\delta}{\omega}\right)$.

Note that for $\delta \ll \omega$:

$$\tan\left(\omega\frac{\tau}{2}\right)\left(\frac{\delta}{\omega}\right) = \tan\left(\frac{\pi}{2\left(1 + \frac{\delta}{\omega}\right)}\right)\left(\frac{\delta}{\omega}\right) \approx \frac{2}{\pi}, \tag{28}$$

which implies:

$$H_{\text{eff}} \approx \left[A\left(\frac{2}{\pi}\right)\cos(\delta t) + B\left(\frac{2}{\pi}\right)\sin(\delta t)\right]\sigma_z \quad (\delta \ll \omega). \tag{29}$$

It should be noted that this is the relevant regime for experimental

realizations[15,34,47,48]. Similarly for the opposite limit $(\delta \gg \omega)$, we obtain that:

$$H_{\text{eff}} \approx \left[ A\left(\frac{\pi}{2}\right)\cos(\delta t) + B\left(\frac{\pi}{2}\right)\sin(\delta t) \right]\sigma_z \quad (\omega \ll \delta). \qquad (30)$$

This derivation can be trivially extended for a signal that consists of two frequencies.

## Data availability

The code and data used in this work are available on request to the corresponding author.

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

## Acknowledgements

The authors thank Konrad Banaszek for helpful discussions. This project has received funding from the European Union Horizon 2020 research and innovation program ERC grant QRES under grant agreement no. 770929 and the collaborative European project ASTERIQS. T.G. is supported by the Adams Fellowship Program of the Israel Academy of Sciences and Humanities.

## Author contributions

A. Retzker and T.G. conceived the idea, the theoretical and numerical analysis was performed by T.G., assisted by A. Rotem. The paper was written by T.G. and A. Retzker. A. Retzker supervised the project.

## Competing interests

The authors declare no competing interests.
