## [Peer Review File · Nature Communications]

Reviewers' Comments:

Reviewer #1:

Remarks to the Author:

The paper discusses the idea of super-resolution in quantum sensing, with a particular focus on NMR models of signal frequencies separation estimation. Authors discuss conditions under which even though the properties of the Hamiltonian make the state formally insensitive to variations of the frequency difference parameter at the point when this parameter is zero, the quantum fisher information is still nonzero. They also discuss an example of noise assisted model where thanks to the presence of fluctuating amplitudes of the signals it is possible to reach the super-resolution limit.

I find the topic analyzed by the authors timely and important and I appreciate their effort to try to provide a general perspective on the super-resolution which abstracts from a particular model. This makes the paper potentially suitable for publication in Nature Communications. Still in its present form, the paper does not convincingly provide such a broad perspective. This is in my opinion mainly due to the structure of the paper.

To be more specific. In Sec IIA authors consider an NMR model where they show that thanks to introduction of a specific noise it is possible to regain the super-resolution. After this example they provide a no go theorem that after superficial reading seems to contradict what was shown in the example before, by claiming that if Hamiltonian does not change under the change of the parameter fisher information is zero "for any possible control and measurement strategy". The reader will surely be confused by this. Of course there is no contradiction, as in the previous example where the "noise" helped to recover the super-resolution it actually broke the symmetry in individual realizations of the Hamiltonian and this was the source of the sensitivity. Still this is very badly explained. Furtheron the authors provide a general conditions that the mixed state need to satisfy in order to allow for super-resolution.

1. For me the order of material is not logical and confusing for the reader. If I understood the paper correctly a much better structure would be along the following lines

- introduction of the frequency separation problem (Eq (1))
- prove Claim 1 (Eq 15) which shows that for pure states and symmetric Hamiltonian there is no possibility for super-resolution
- then discuss the mixed state case, and show that this allows in principle to overcome this "pure state claim".
- then show the noise-assited protocol which actually do exactly this - transfers us from the pure state model to a mixed state model, where despite the symmetry of the state we have super-resolution.

2. Apart from restructuring suggestion, my second main issue is the fact that instead of treating the problem in greater generality the reader is mainly faced with the NMR frequency separation model. There is a discussion of optical superresolution imaging that shows the broader applicability of the results, but the paper lacks elegant general mathematical formulation from the start that would allow

the reader to see the NMR and optical problems as special cases of a general framework. Such a formulation would make this paper much more elegant and convincing for the reader as it would combine in a single formalism many apparently different physical scenarios.

3. The only imperfection the authors consider is the mismatch in the estimation of the central frequency (authors might refer at this point to analogous consideration in optical super-resolved imaging: see e.g. arXiv:1512.08304, arXiv:1803.07096). In my opinion in order not to leave the reader with overidealized picture, authors should at least mention the impact of some simple typical noise, like e.g. dephasing noise. The noise that authors consider in their example is a very specific noise with long memory and hence allowing to perform very precise sensing. In contrast typical Markovian noises are usually quite unforgiving, and it would be illuminating if authors tried at least to provide a basic discussion using e.g. the general tools developed in arXiv:1704.06280, arXiv:1706.02445, where the impact of Markovian noise on sensing capabilities may be taken into account. In particular, how fluctuations of phase or estimated frequencies affect the picture. I would expect it imposes some additional limit on the regime where super-resolution may be observed.

To sum up, the paper in my opinion is a promising candidate for a Nature Communication publication but requires some additional work in order to better highlight its "model independent" content, help the reader to follow the logic of the reasoning, and lead to a joined perspective on problems where super-resolution appears in different physical contexts.

Reviewer #2:

Remarks to the Author:

In this paper, the authors mainly studied quantum metrology for the frequencies of two oscillating signals in a qubit Hamiltonian. The authors used pi-pulses to increase the resolution limit and show that projective measurements which nullify the measurement noises can help obtain non-zero Fisher information. The authors further showed a no-go theorem for the polarized NMR resolution, and gave a general criterion about frequency resolution for mixed quantum states.

The paper includes some interesting result, but I did not find them innovative enough to warrant publication in Nature Communications. Below are my main concerns with the this paper.

1. The authors used quantum control to increase the Fisher information, but such control method has been used in many recent papers, so it seems that the authors just applied quantum control to the NMR model, which is more a case study than a significant innovation.

2. The authors showed that the Fisher information for ω_r is zero when the Hamiltonian is symmetric about ω_r and $-\omega_r$ at $\omega_r=0$. The result is mathematically correct. But for this case, in the original model (1), ω_1 and ω_2 plays the same role which can be interchanged, so in fact, it does not matter which is larger. So, in this case, it seems to me that it is more meaningful to estimate $|\omega_r|$ rather than ω_r . And the Fisher information for $|\omega_r|$ will not be zero.

3. The authors showed that pi-pulses can significantly increase the Fisher information for ω_r . The method is to choose the right δ_s so that $\delta_s t = 2\pi$, and the pi-pulses is applied at a frequency $\omega + \delta$, which should be $\omega_1 + \delta_1$ and $\omega_2 + \delta_2$ here as there are

two frequencies. However, ω_1 and ω_2 are unknown and need to be measured. So, how can this control method work in practice?

In addition, A_i and B_i are random, so how can we apply the π -pulses which involves A_i and B_i ?

Based on the above comments, I think the current paper needs revision and is suitable for more specific journals than Nature Communications.

Reply - Overcoming resolution limit with quantum sensing

Referee 1

Referee: I find the topic analyzed by the authors timely and important and I appreciate their effort to try to provide a general perspective on the super-resolution which abstracts from a particular model. This makes the paper potentially suitable for publication in Nature Communications. Still in its present form, the paper does not convincingly provide such a broad perspective. This is in my opinion mainly due to the structure of the paper.

To be more specific. In Sec IIA authors consider an NMR model where they show that thanks to introduction of a specific noise it is possible to regain the super-resolution. After this example they provide a no go theorem that after superficial reading seems to contradict what was shown in the example before, by claiming that if Hamiltonian does not change under the change of the parameter fisher information is zero "for any possible control and measurement strategy". The reader will surely be confused by this. Of course there is no contradiction, as in the previous example where the "noise" helped to recover the super-resolution it actually broke the symmetry in individual realizations of the Hamiltonian and this was the source of the sensitivity. Still this is very badly explained. Furtheron the authors provide a general conditions that the mixed state need to satisfy in order to allow for super-resolution.

1. For me the order of material is not logical and confusing for the reader. If I understood the paper correctly a much better structure would be along the following lines

- introduction of the frequency separation problem (Eq (1))
- prove Claim 1 (Eq 15) which shows that for pure states and symmetric Hamiltonian there is no possibility for super-resolution
- then discuss the mixed state case, and show that this allows in principle to overcome this "pure state claim".
- then show the noise-assisted protocol which actually do exactly this - transfers us from the pure state model to a mixed state model, where despite the symmetry of the state we have super-resolution.

Reply: We thank the referee for the helpful feedback on restructuring of the paper, which we have implemented (partly because of other responses we received: some people misinterpreted this paper as just finding optimal pulse sequence for a specific problem). We therefore changed the structure accordingly. The structure of the results section now reads:

Subsection A: Conditions for superresolution. In this section we:

1. Formulate quantum spectroscopy problems, and illustrate why a limitation appears in the case of pure state.
2. Move on to the general picture: vanishing derivative of the density matrix.
3. statement and proof of the general condition.

(to make it more general we also added a multivariate condition).

Subsection B presents the main application, which we term resolution without coherence.

Subsection C: Limitations and imperfections of this method (we address this section later in the report)

Subsection D: Additional applications-quantum resolution methods in sampling. This section is new and it addresses the case in which the coherence time of the signal is much longer than the coherence time of the probe. We present two methods: superresolution with quantum Fourier transform and superresolution with correlation spectroscopy. Both methods are special cases of the condition presented in section A.

Referee: Apart from restructuring suggestion, my second main issue is the fact that instead of treating the problem in greater generality the reader is mainly faced with the NMR frequency separation model. There is a discussion of optical superresolution imaging that shows the broader applicability of the results, but the paper lacks elegant general mathematical formulation from the start that would allow the reader to see the NMR and optical problems as special cases of a general framework. Such a formulation would make this paper much more elegant and convincing for the reader as it would combine in a single formalism many apparently different physical scenarios.

Reply: We thank the referee for this comment, we generalized the results as much as possible, and the paper starts now with this generalization. In subsection A we define vanishing distinguishability, and try to convince the readers that in general, resolution problems suffer from this property. We then formulate a general condition to overcome this, and clarify that the recent quantum resolution ideas in optical imaging are special cases of this criterion.

Note that two new methods that are based on this criterion were added to the paper. We explicitly write that all these methods are special cases of the criterion presented in subsection A.

Referee: The only imperfection the authors consider is the mismatch in the estimation of the central frequency (authors might refer at this point to analogous consideration in optical super-resolved imaging: see e.g arXiv:1512.08304, arXiv:1803.07096). In my opinion in order not to leave the reader with overidealized picture, authors should at least mention the impact of some simple typical noise, like e.g. dephasing noise. The noise that authors consider in their example is a very specific noise with long memory and hence allowing to perform very precise sensing. In contrast typical Markovian noises are usually quite unforgiving, and it would be illuminating if authors tried at least to provide a basic discussion using e.g. the general tools developed in arXiv:1704.06280, arXiv:1706.02445, where the impact of Markovian noise on sensing capabilities may be taken into account. In particular, how fluctuations of phase or estimated frequencies affect the picture. I would expect it imposes some additional limit on the regime where super-resolution may be observed.

Reply: We thank the referee for this comment. We analyzed two relevant and important imperfections, and indeed realized that they impose further limitations. These imperfections are:

1. (Markovian) Dephasing of the probe. We choose to analyze the most harmful noise type, i.e., when the noise and the signal are “parallel” (the jump operator in the master equation and the Hamiltonian are linear dependent).

2. Fluctuations of the amplitudes/phases of the signal (incoherence of the signal during the measurement).

The limitations imposed by these factors are briefly analyzed in the main text (section C: “incoherence of the signal”, “dephasing of the probe”), and a detailed analysis is presented in the supplemental (section 7-”Imitation due to incoherence”, section 8-”Limitation due to dephasing of the probe”). We summarize the results here:

Limitations due to dephasing of the probe: Dephasing of the probe inflicts a limitation in a the following way. Introducing a dephasing to the master equation (with a dephasing rate of κ), leads to the following transition probability:

$$p = 0.5 \left(1 - \exp \left(- \frac{(\sigma t)^2 (\omega_s t)^2}{\pi^2} - 2\kappa t \right) \right).$$

So there is a competition between $(\sigma t)^2 (\omega_s t)^2$ and κt , resolution is thus achieved only if the first term is larger. Therefore resolution is achieved only for: $\frac{\omega_s \sigma}{\kappa^2} \gg 1$.

This inequality sets a limitation on the strength of the signal. This limitation is not crucial in most cases as usually the coherence of the probe doesn’t set the bottleneck for resolution, the bottleneck is typically set by the coherence of the signal. Namely if the coherence time of the probe is very short (relative to the coherence time of the signal), you would prefer to sample the signal via synchronized Ramsey measurements, as it is done in (for example) Science 356.6340 (2017): 832-837, Science 356.6340 (2017): 837-840.

The referee mentions the following papers: arXiv:1704.06280, arXiv:1706.02445, which set conditions for achieving the Heisenberg limit in quantum metrology. We didn’t make a rigorous analysis of how quantum error correction can improve our method, however we point out in the supplemental that error correction can be useful in our method even in cases where it is not useful according the papers mentioned above. The paragraph about this reads:

“We therefore observe that dephasing imposes a resolution limit, this invokes a natural question: can error correction protocols remove limitations imposed by a Markovian noise? Note that this question is not analogous to the achievability of Heisenberg scaling which was addressed in [arXiv:1704.06280, arXiv:1706.02445]. The prospects for quantum error correction in these resolution problems are left for future work, however we point out that this limit can be eliminated provided that errors can be detected, since one does not need to correct the errors. Given that error detection is possible, we can postselect the measurements without error and perform estimation according to them. These errors thus reduce the precision, but they do not impose a limitation. This implies that the error correction condition to remove limitation in this problem should be different from the error correction condition for Heisenberg scaling. Let us denote the jump operators in the Master equation as $\{L_j\}_j$.

For Heisenberg limit, the error correction condition is $H \notin \text{span} \left\{ I, L_j, L_j^\dagger, L_i^\dagger L_j \right\}_{i,j}$ [arXiv:1704.06280, arXiv:1706.02445], since we need both to detect and to correct. In this case, since we need only to detect, the error correction condition should be: $H \notin \text{span} \left\{ I, L_j^\dagger, L_j \right\}_j$. This implies, for example, that if the only noise source is amplitude damping (namely the only jump operator is σ_-), then detection is possible, and we can overcome the limitation. “

Limitations due to incoherence of the signal: We assumed that the amplitudes undergo Orenstein - Uhlenbeck (OU) noise process. Given this noise process, the accumulated phase is not going to vanish:

Due to these fluctuations the transition probability includes an additional noise term(denoted as n), such that it reads (for $\omega_s t =$

2π):

$$p = \frac{\sigma^2 t^2}{2\pi^2} \omega_r^2 t^2 + n.$$

This new term imposes a limitation: It is now impossible to nullify p , which implies that $I_r \rightarrow 0$ as $\omega_r \rightarrow 0$. Resolution can be achieved only if the first term is larger than n . Taking into account OU process: it is characterized by a decay rate (γ) and a standard deviation (σ_n). Given these parameters the transition probability reads:

$$p \approx \frac{\sigma_n^2 T^3}{\pi^2} + \frac{\sigma_n^2}{4\pi^2 \gamma} \omega_r^2 T^4,$$

where the first term is n and the second term is the original transition probability that gives us information about ω_r . Comparing these two terms, it can be seen that resolution can be achieved only for $\omega_r/\gamma > 1$.

This indeed limits the applicability of this method. It means that if you lose coherence only due to these fluctuations, you cannot gain using this method. Hence this method is relevant mainly for shot-to-shot noise, namely when the signal is relatively coherent in each measurement but the amplitudes vary between measurements. This type of noise, however, is ubiquitous in quantum optical systems as the preparation / measurement time is usually much longer than the probing time and the time separation between two consecutive measurements is much longer than the correlation time of the noise. Here are some possible implementations: 1. In some cases the readout process of the probe takes a lot of time (such as in single-shot readout of the NV). In these cases the noise term during each phase acquisition can be very small, however until you finish the readout the signal already lost its coherence. This effectively creates a shot-to-shot noise. 2. It can be used for communication: you can send coherent yet very short signals, each signal with random amplitudes. The information is encoded in ω_r , and then one can retrieve this information only by using this method.

Of course there are maybe a lot of other instances (that we are not aware of) in which this method is useful.

Referee 2

Referee: The paper includes some interesting result, but I did not find them innovative enough to warrant publication in Nature Communications. Below are my main concerns with the this paper.

1. The authors used quantum control to increase the Fisher information, but such control method has been used in many recent papers, so it seems that the authors just applied quantum control to the NMR model, which is more a case study than a significant innovation.

Reply: We respectfully disagree. In this paper we introduce a novel quantum super-resolution protocol which eliminates resolution issues in a scenario which was previously believed to be unresolvable. In order to do that we use a few ingredients and indeed control is one of them. We of course agree that control methods are routinely used in quantum sensing and in nano - NMR in particular. In fact, control methods are incorporated in almost all nano - NMR experiments, except the microwave free measurements. However, the fact that control was also used in other contributions does not reduce the importance of our findings as control is not the main issue here. This paper is not about increasing the FI via control methods, but about eliminating the divergence of the estimation variance in resolution problems, using quantum features.

This project indeed started as a case study: we just wanted to calculate the FI of ω_r for the “resolution without coherence” problem. We were quite confident that the FI vanishes in the limit of $\omega_r \rightarrow 0$, because of the vanishing distinguishability property: $\frac{dp}{d\omega_r} \rightarrow 0$ as $\omega_r \rightarrow 0$ (for any control method/ measurement basis).

But to our surprise we observed that very specific control methods (in fact, a zero measure set of them) achieve a non-vanishing FI. These are the methods that nullify projection noise.

It was then realized that this observation is quite general and relevant for other problems as well. In fact, a very similar idea is used in Tsang’s optical superresolution papers. This inspired us to formulate a general superresolution condition (for single variable and for multivariable case), that summarizes what are the cases in which superresolution can be achieved.

Hence the main idea of the paper (and the main message we try to convey) is illustrated in figure 1:

In the current version, we present two additional superresolution methods that use this generic principle. The first method uses quantum Fourier transform (QFT). It is known that if we make measurements and perform a classical Fourier transform, then we suffer from a resolution limit. We show that if we store the data (the phases) in a quantum state and measure it in the Fourier basis we eliminate the resolution limit. Afterwards we show that using a specific kind of correlation spectroscopy method, one can also eliminate the resolution limit and achieve similar results.

Therefore we believe that these ideas are important and fundamental to a broad range of audience. In fact, we have been asked several times by colleagues (from different quantum related fields) if there is a relation between resolution problems

Figure 1: The main idea of the paper

in spectroscopy and resolution problems in optical imaging, and whether superresolution ideas from optical imaging can be implemented in spectroscopy. We provide in this paper an (affirmative) answer, and show that ideas that work in optical imaging can be translated to spectroscopy, and we formulate a condition that generalizes all the relevant methods.

Referee: 2. The authors showed that the Fisher information for ω_r is zero when the Hamiltonian is symmetric about ω_r and $-\omega_r$ at $\omega_r = 0$. The result is mathematically correct. But for this case, in the original model (1), ω_1 and ω_2 plays the same role which can be interchanged, so in fact, it does not matter which is larger. So, in this case, it seems to me that it is more meaningful to estimate $|\omega_r|$ rather than ω_r . And the Fisher information for $|\omega_r|$ will not be zero.

Reply: We agree that it is sufficient to estimate $|\omega_r|$, and in fact when we simulated the estimation procedure we (in practice) estimated $|\omega_r|$, as you suggest (the probability is symmetric with respect to ω_r , so you can't really differentiate between ω_r and $-\omega_r$. In practice there is no difference between ω_r and $-\omega_r$). You can see in the results that there is no difference between estimation of ω_r and estimation of $|\omega_r|$.

The FI about $|\omega_r|$ is exactly the same as the FI about ω_r , and therefore it also vanishes. To see this observe that:

$$\lim_{\omega_r \rightarrow 0} \frac{dp}{d\omega_r} = 0 \Rightarrow \lim_{\omega_r \rightarrow 0^+} \frac{dp}{d\omega_r} = \lim_{\omega_r \rightarrow 0^-} \frac{dp}{d\omega_r} = 0$$

But:

$$\lim_{\omega_r \rightarrow 0^+} \frac{dp}{d\omega_r} = \lim_{\omega_r \rightarrow 0^-} \frac{dp}{d\omega_r} = 0 \Rightarrow \lim_{|\omega_r| \rightarrow 0} \frac{dp}{d|\omega_r|} = \lim_{\omega_r \rightarrow 0^+} \frac{dp}{d\omega_r} = \lim_{\omega_r \rightarrow 0^-} \frac{dp}{d\omega_r} = 0.$$

So the the FI about ω_r coincides with the FI about $|\omega_r|$, and therefore the FI about $|\omega_r|$ vanishes as well.

Referee: 3. The authors showed that pi-pulses can significantly increase the Fisher information for ω_r . The method is to choose the right δ_s so that $\delta_s * t = 2 * \pi i$, and the pi-pulses is applied at a frequency $\omega_r + \delta_s$, which should be $\omega_1 + \delta_1$ and $\omega_2 + \delta_2$ here as there are two frequencies. However, ω_1 and ω_2 are unknown and need to be measured. So, how can this control method work in practice? In addition, A_i and B_i are random, so how can we apply the pi-pulses which involves A_i and B_i ?

Reply: The frequency of the π -pulses indeed depends on $\omega_s = 0.5(\omega_1 + \omega_2)$. A non-vanishing FI is achieved only when the frequency of the π -pulses is $\omega_s + \frac{2\pi n}{T}$ (for an integer n). This fact is mentioned several times in the paper. The solution to this alleged problem is explained in detail in the paper: ω_s can be estimated efficiently, prior to applying this method.

This point is addressed in the main text, section C (referred to as “Unknown parameters”), quote it here:

“Consider now the case in which σ and ω_s are unknown. Since the estimation protocol of ω_r depends on knowledge of ω_s , we must first estimate ω_s . This can be done using the traditional method : Applying π -pulses in different frequencies and fitting the transition probability as a function of the pulses frequency. This should provide a good estimation of σ, ω_s , but not a good enough estimation of ω_r (unless by chance we hit close enough to a resonance frequency). Once a good enough estimation of ω_s is obtained we can apply the required control ($\delta_s t = 2\pi$). To understand what is a good enough estimation of ω_s observe that for small enough $\omega_r t, (\delta_s - 2\pi) t: I_r = \frac{8\sigma^2 t^4}{\pi^4} \frac{\omega_r^2}{\omega_r^2 + (\delta_s - 2\pi)^2}$, hence the width of the resonance peak (in $\delta_s t$) goes as $\omega_r t$. Therefore once $\Delta\omega_s$ is comparable to ω_r this method works despite the small detuning.

Observe that now a multivariate estimation should be performed, which means that at least three different measurements are needed; each measurement in a detuning that is optimal for a different parameter. Numerical results are shown in the supplemental.”

An elaborate analysis is presented in section 6 of the supplemental (“Multiparameter estimation of all the parameters”). In this section we present numerical results, and show how to perform an initial estimation of ω_s and then use it for our method.

Regarding the question about A_i, B_i : note that the control method does not depend on A_i, B_i . As long as $\delta_s t = 2\pi$, the transition probability is nullified for any values of A_i, B_i . The only important assumption about the distribution of A_i, B_i is that A_1, B_1 are not correlated with A_2, B_2 . this method for example would not work if A_1, B_1 are random, but $A_2 = A_1, B_2 = B_1$ (because then the transition prob. would go as ω_r^4).

Reviewers' Comments:

Reviewer #1:

Remarks to the Author:

The authors have addressed my main concern, and indeed the paper has now much better structure and allows the reader to get a bigger picture view on the problem. I also appreciate extended discussion on the impact of noise and multiparameter case.

I just have some minor technical remarks that came to my mind on the second reading of the manuscript:

1. Concerning the main claim, just above Eq.3. The authors argue that the only possibility is the scaling of an eigenvalue as ω_r^2 . It seems to me that any power $1 < k \leq 2$ would do. Of course for $1 < r < 2$ we would have a divergent FI, but this is a manifestation of "extreme superresolution potential". Since authors do not consider any specific encoding model for the parameter they should take into account also this possibility or at least comment on it.

2. Above Eq.(2) when authors discuss a general multiparameter case, they should write $g \rightarrow 0$ rather than $\omega_r \rightarrow 0$.

I think the paper is now suitable for publication in Nature Communications.

Reviewer #2:

Remarks to the Author:

I have read the revised manuscript carefully, and I do not think the paper is suitable for publication in Nature Communications. The reason is as follows.

1. The paper focuses on resolving the separation of two frequencies, assuming that the center of the two frequencies can be estimated by some fitting method. This actually oversimplifies the problem. After all, the purpose of frequency spectroscopy is to determine the participating frequencies, not only the relative separation between them. If only the separation of the frequencies has a good resolution, but the center of the frequencies does not, the result of spectroscopy will still be low-precision. So, the frequency spectroscopy is always a multi-parameter estimation problem.

I noticed the authors discussed this issue in the subsection "Unknown parameters". But I do not find their claim convincing. The authors claimed that the traditional method by fitting transition probability as a function of frequencies can give a good estimate for the center of the frequencies (as well as for the variance of quadrature A and B). The question is how good it is compared to the precision of the separation of two frequencies? If the resolution of the center of frequencies is not as high as the separation of the frequencies, it will be senseless to resolve the separation of the two frequencies, as the precision of two frequencies is finally determined by both their center and separation. If we want to have a high resolution for the center of frequencies as well, then specific quantum measurements need to be designed for it, and the unknown separation of two frequencies needs to be measured at the same time. Therefore, the separation and the center of two frequencies must be measured simultaneously, the frequency resolution problem cannot be simplified to a single-parameter estimation problem as in the current paper.

2. In addition, I think about the main result Eq. (7) more carefully in reviewing the revised manuscript, and it seems a little problematic. To derive Eq. (7), the authors averaged the accumulated phase in Eq. (6) due to fluctuation of A_i and B_i , and used it to compute the Fisher information I_r . Although one can average the phase as in (6), but for the current model in the paper, the qubit evolves in a pure state for each realization of A_i and B_i . So, shouldn't the Fisher information be computed for each realization first, then be averaged over all possible realizations? The current result seems not exact.

3. Still about the result (7), the scaling of I_r is not correct. The authors replaced ω_s by $2\pi n/t$ in order to have a higher time scaling. But when t increases, n also increases, and the n^2 in the denominator scaling will decrease I_r by t^2 . So, the remained net time scaling of I_r is only t^2 , which is the just traditional scaling. The same issue exists in the result (11).

Reply - Overcoming resolution limit with quantum sensing

Referee 1

Referee: Concerning the main claim, just above Eq.3. The authors argue that the only possibility is the scaling of an eigenvalue as ω_r^2 . It seems to me that any power $1 < k \leq 2$ would do. Of course for $1 < r < 2$ we would have a divergent FI, but this is a manifestation of "extreme superresolution potential". Since authors do not consider any specific encoding model for the parameter they should take into account also this possibility or at least comment on it.

Reply: The referee is right, the reason we mentioned only ω_r^2 is because we assumed that the eigenvalues/probabilities are analytic, and because we don't know of any example in which $p \sim \omega_r^k$ for $1 < k < 2$, or in general any example in quantum sensing in which the FI diverges (where the dimension of the Hilbert space is finite). However we cannot rule out this option and agree that the most general case should be considered, so we changed accordingly. The claim now reads:

"Given $\rho(\omega_r)$ such that $\frac{d\rho}{d\omega_r} = 0$ (as $\omega_r \rightarrow 0$), then $I_r > 0$ if and only if at least one of the eigenvalues of ρ goes as ω_r^k , where $1 < k \leq 2$ or equivalently $\frac{d\sqrt{\rho}}{d\omega_r} \neq 0$. The optimal measurement basis converges to an eigenbasis of ρ as $\omega_r \rightarrow 0$."

We mention in a footnote that any $1 < k < 2$ leads to a divergent FI.

Interestingly, although we can mathematically write a density matrix that behaves like this and thus achieves a divergent QFI, we don't know of any physical realistic model that yields such a behavior.

Referee: Above Eq.(2) when authors discuss a general multiparameter case, they should write $g \rightarrow 0$ rather than $\omega_r \rightarrow 0$.

Reply: We thank the referee for this comment, we are afraid there is maybe a slight misunderstanding here so let us first clarify the original sentence ("Indeed, in many resolution problems there exists g such that $\frac{\partial \rho}{\partial g} \rightarrow 0$ as $\omega_r \rightarrow 0$.") :

When talking about resolution, we always take the limit of vanishing separation parameter $\omega_r \rightarrow 0$ (or in optics $d \rightarrow 0$). In this limit, resolution limit appears when the derivative according to a certain parameter vanishes, however this problematic parameter is not necessarily ω_r , and thus we denote it as g . Consider for example the case of a signal with different amplitudes (and identical phases), the problematic parameter would be $\omega_- = \frac{1}{\sqrt{\Omega_1^2 + \Omega_2^2}} (\Omega_2 \omega_1 - \Omega_1 \omega_2)$ (see supplemental section 3). The problematic parameter is ω_r when the problem is symmetric, i.e. there is a symmetry of swapping the two frequencies/sources, like in our case.

We think that the referee probably thought that g is a more general notation of the separation parameter, however we keep the same notation throughout the paper: ω_r is the separation parameter. In order to make this sentence clearer, we changed it to:

"Indeed, in many resolution problems as we take the separation parameter ω_r (the difference between the frequencies or, in imaging, the sources) to 0, there exists a parameter g such that $\frac{\partial \rho}{\partial g} = 0$."

We hope this answers the referee's comment.

Referee 2

Referee: The paper focuses on resolving the separation of two frequencies, assuming that the center of the two frequencies can be estimated by some fitting method. This actually oversimplifies the problem. After all, the purpose of frequency spectroscopy is to determine the participating frequencies, not only the relative separation between them. If only the separation of the frequencies has a good resolution, but the center of the frequencies does not, the result of spectroscopy will still be low-precision. So, the frequency spectroscopy is always a multi-parameter estimation problem.

I noticed the authors discussed this issue in the subsection Unknown parameters. But I do not find their claim convincing. The authors claimed that the traditional method by fitting transition probability as a function of frequencies can give a good estimate for the center of the frequencies (as well as for the variance of quadrature A and B). The question is how good it is compared to the precision of the separation of two frequencies? If the resolution of the center of frequencies is not as high as the separation of the frequencies, it will be senseless to resolve the separation of the two frequencies, as the precision of two frequencies is finally determined by both their center and separation. If we want to have a high resolution for the center of frequencies as well, then specific quantum measurements need to be designed for it, and the unknown separation of two frequencies needs to be measured at the same time. Therefore, the separation and the center of two frequencies must be measured simultaneously, the frequency

resolution problem cannot be simplified to a single-parameter estimation problem as in the current paper.

Reply: The main point in the objection presented above is:

”If we want to have a high resolution for the center of frequencies as well, then specific quantum measurements need to be designed for it, and the unknown separation of two frequencies needs to be measured at the same time. Therefore, the separation and the center of two frequencies must be measured simultaneously, the frequency resolution problem cannot be simplified to a single-parameter estimation problem as in the current paper.”

However, we have made exactly this analysis in the supplemental. We assume that this analysis was missed by the referee as it is presented in the supplemental (section 6) and not in the main text.

We agree, of course, that spectroscopy is a multi-parameter estimation problem. Essentially ω_1, ω_2 need to be estimated, or equivalently ω_s, ω_r (and indeed divergence of $\Delta\omega_s$ would also be a problem).

1. As the referee suggests we have made a complete multivariate analysis already in the original version of the paper, which is presented in section 6 of the supplemental material. The histograms of the estimators of ω_s, ω_r are presented there, and it is shown that by using our method the precision in estimating ω_s, ω_r are comparable, and resolution is indeed achieved. For convenience we add this figure also here, we added the histograms of ω_1, ω_2 to show that these frequencies are efficiently estimated and resolved (see fig. 1).

We remark that in order to extract information about ω_s, σ we use measurements with different detunings. One cannot gain information about three different parameters by applying the same measurement on a qubit (this will give you information only about a single parameter). Hence after the initial fit, we make measurements in three different detunings to extract information about all the parameters, where only $\delta = 2\pi/\omega_s$ give us information about ω_r . This fact is mentioned also in the main text: “Observe that now a multivariate estimation should be performed, which means that at least three different measurements are needed; each measurement in a detuning that is optimal for a different parameter. Numerical results and further analysis are presented in sec. 6 of the supplemental. ”

Still, you may ask why in the analysis we reduced the problem to a single parameter estimation problem, focusing on I_r , the FI about ω_r ?

2. In subsection A of the results we justify this. In this part we formulate conditions for superresolution, and address the multivariate case: Given that $\left\{ \frac{\partial \rho}{\partial \theta_j} \right\}_j$ is linear dependent, what are conditions for a regular QFI? The condition for the multivariate case reads: “we can choose $(\theta_i)_{i=1}^n$ such that $\left(\frac{\partial \rho}{\partial \theta_i} \right)_{i=1}^k$ are linearly independent and $\frac{\partial \rho}{\partial \theta_{k+1}} = \dots = \frac{\partial \rho}{\partial \theta_n} = 0$ ($\theta_{k+1}, \dots, \theta_n$ are the problematic parameters). Then the QFI is regular if and only if the classical FI matrix (i.e. the FI matrix obtained when measuring in the eigenbasis of ρ) about the problematic parameters $(\theta_{k+1}, \dots, \theta_n)$ is regular.” Namely in general you need to focus only on the problematic parameters; i.e. $\{\theta_k\}_k$ that satisfy $\frac{\partial \rho}{\partial \theta_k} = 0$. There shouldn’t be any problem to estimate the other (non problematic) parameters.

In our problem, fundamentally $\frac{\partial \rho}{\partial \omega_r} = 0$, however $\frac{\partial \rho}{\partial \omega_s}$ doesn’t necessarily vanish. Therefore in principle it is enough to check in which cases the classical FI about ω_r doesn’t vanish. This would guarantee a finite $\Delta\omega_r, \Delta\omega_s$.

Referee: In addition, I think about the main result Eq. (7) more carefully in reviewing the revised manuscript, and it seem a little problematic. To derive Eq. (7), the authors averaged the accumulated phase in Eq. (6) due to fluctuation of A_i and B_i , and used it to compute the Fisher information I_r . Although one can average the phase as in (6), but for the current model in the paper, the qubit evolves in a pure state for each realization of A_i and B_i . So, shouldn’t the Fisher information be computed for each realization first, then be averaged over all possible realizations? The current result seems not exact.

Reply:

We agree that in each realization of A_i, B_i the state is pure, and the measurement is performed on a pure state. However the A_i, B_i in each measurement are not known a priori, as we get a completely random and unknown amplitudes (note that if you would have known the amplitudes in advance, then there wouldn’t be any resolution problem, because you would have known that there are two sets of amplitudes, and thus two frequencies). In short, since you don’t know the amplitudes and they vary between consecutive measurements an averaging of the quantum state should be performed, and thus the relevant quantum state is the (averaged) density matrix and the relevant figure of merit is the FI computed according to this density matrix. The quantity that the referee suggest, $\langle I_r \rangle_{A_i, B_i}$, is not very meaningful in this problem, it’s the FI that you would have in case that the amplitudes are known in advance. Indeed if all the amplitudes are known, then there is no resolution problem from the first place.

In principle, you can argue that we can make many measurements in each realization, and try to estimate all the parameters (frequencies, amplitudes), and then there is no need to average the quantum state. However it was shown in PRL, 122(6), 060503 (“Limits on spectral resolution...”) that also in this case you get a divergence. Only if you repeat this process for many different realizations and perform a very specific (and quite involved) post processing you can eliminate the divergence.

Figure 1: The second stage of the protocol consists of measurements in three different detunings (each one is optimal for the estimation of a different parameter; one of them is therefore $\delta_s = \frac{2\pi}{t}$). In this example: $\delta_s t = 0.01$, $\sigma t = 5$ and the number of measurements in each detuning is $N = 3 \cdot 10^5$. The histograms for the different parameters are presented in (a). It can be seen that resolution is achieved as $\Delta\omega_r = \frac{1}{7.5}\omega_r$. In (b) the histograms of ω_1, ω_2 are presented, it can be seen that the frequencies are clearly resolved. (c) The RMSE (root mean square error) in estimating the parameters as a function of N . The solid lines are the theoretical expectations (green (bottom): $\Delta\omega_s$, orange (middle): $\Delta\omega_r$, blue (top): $\Delta\sigma$). The points represent the RMSE achieved in an actual maximum likelihood estimation (green (diamonds): $\Delta\omega_s$, orange (circles): $\Delta\omega_r$, blue (squares): $\Delta\sigma$). It can be seen that there is no divergence: The RMSE of all the parameters scale as $N^{-0.5}$. In fact $\Delta\omega_r$ is very close to $\Delta\omega_s$, whereas the worst is $\Delta\sigma$.

Furthermore in this paper we investigated a more challenging regime, a regime in which each realization is accessed only once. Namely due to the varying amplitudes you can't make more than a single measurement in each realization. The fact that resolution can be achieved in this case surprised us a lot.

To clarify and summarize: when dealing with noise the Fisher Information has to be calculated according to the density matrix and not from the pure states comprising its purification. Doing it the other way around could lead to nonsensical results. For example: suppose each measurement is performed on a single realization of (a completely random phase ϕ):

$$\rho = \frac{1}{2\pi} \int_0^{2\pi} |\uparrow e^{i(g+\phi)} + \downarrow e^{-i(g+\phi)}\rangle \langle \uparrow e^{i(g+\phi)} + \downarrow e^{-i(g+\phi)}| d\phi = \mathbb{I} \quad (1)$$

The information about g has to vanish, there is no way to estimate g given that the phase (ϕ) is completely random. Indeed The Fisher Information of the identity density matrix is of course zero. However, if we calculate the Fisher information first and then average we get:

$$FI = \frac{1}{2\pi} \int_0^{2\pi} 4 d\phi = 4, \quad (2)$$

which is of course wrong.

Referee: Still about the result (7), the scaling of I_r is not correct. The authors replaced ω_s by $2\pi n/t$ in order to have a higher time scaling. But when t increases, n also increases, and the n^2 in the denominator scaling will decrease I_r by t^2 . So, the remained net time scaling of I_r is only t^2 , which is the just traditional scaling. The same issue exists in the result (11).

Reply: We agree with the referee that in eq. (7) the scaling with time can be misleading (the formula is correct, but in principle replacing ω_s with n/t is misleading), we therefore changed it. However we never meant to leave the reader with the impression that a very good FI is achieved in this case (as the referee suggests), we stress the fact that this FI can be very poor and control is needed to improve it. We wrote it this way in order to explicitly show the effect of n , the number of periods completed during the measurement. Anyway, we changed it, and it now reads:

“Hence the superresolution condition is satisfied and the FI reads:

$$I_r = \frac{8\sigma^2 t^2}{\omega_s^2}. \quad (3)$$

...

The obtained FI can be still quite poor and far from optimal. Note that it goes as $1/n^2$, where n is the number of periods completed during the measurement. If n is large, then this factor of $\frac{1}{n^2}$ can be significant. A much better FI can be achieved by applying a suitable control: π -pulses which effectively change the frequency of oscillations, and reduce n to 1. ”

We respectfully disagree about eq. 11, in this equation the scaling is accurate. The idea we wanted to show is that by applying control, we can always reduce n to 1. Note that as t gets larger the coherent control is modified, and the effective frequency gets smaller ($\delta_s = 2\pi/t$), hence in this formula there is no catch, the scaling is indeed t^4 . That’s true also for a single frequency case: the optimal strategy is to reduce δ to 0 (with an appropriate initial phase/ timing of pulses), but if instead we apply a control that takes δ to π/t (for every t), the scaling is still t^4 but we will lose a factor (which is independent of t).

Reviewers' Comments:

Reviewer #1:

Remarks to the Author:

I fully support Authors' answers to the second Referee. In my opinion they satisfactory address all the issues raised. I can only reiterate my recommendation for publication.

Reviewer #2:

Remarks to the Author:

I have read the author's revised manuscript. I think the authors did not catch the main points of the comments in my last report.

1. I noticed that the authors made a multivariate analysis for ω_s and ω_r in the supplemental material. But a simple histogram is not a sufficient analysis. The core is how to most efficiently extract the information of the two parameters at the same time. As I said in the last report, the authors mainly focused on the estimation of ω_r , but why can ω_s be neglected after the initial ?

I understand that the purpose of this work is to enhance the distinguishability of two frequencies, but the analysis should be put on the framework of multivariate estimation, otherwise it would be senseless. In multivariate parameter estimation, the precision (or Fisher information) of different parameters are often complementary (Nat. Commun. 5, 3532 (2014) showed a very good example for phase and phase diffusion), so in the current problem, the precision of ω_s may influence the precision of ω_r . How the precision of these two parameters interplay with each other is critical to the simultaneous estimation of these parameters, and needs to be investigated in detail.

In fact, when the authors did a rough fit for the parameter ω_s initially, they indeed did it for all unknown parameters ω_s , ω_r and σ . So why can the authors change to focus on ω_r alone after the initial estimation? The authors gave a reason in the reply as well as in the main text that the QFI is regular. This is far from sufficient. There are two issues here: (i) The regularity of QFI does not mean the other parameters do not have influence on ω_r , it just means that it is possible to estimate all parameters; how to optimally estimate these parameters is still unknown. (ii) The QFI is not always saturable. There is a very strict condition for the saturability of QFI. A detailed analysis about this problem can be found in Phys. Rev. A 94, 052108 (2016). In addition, I also want to refer to the original superresolution paper Phys. Rev. X 6, 031033 (2016) which treats this problem in an exact multivariate way.

2. For the problem of average over realizations of A_i and B_i , even there is only one measurement for each realization, the Fisher information should still be averaged over all possible realizations. The reason is whatever A_i and B_i occurs in reality for each measurement, they are always random, and the measurement results are also random, so the deviation of the estimator is random as well and needs to be averaged over the realizations and the measurement results. The average over measurement results will give Fisher information, and the average over realizations will give the average Fisher information which I mentioned in the last report. Averaging over the qubit state does not give any physical things, because this is not the state of the qubit in reality for any realization.

The authors gave an example that the average of a single realization with a random phase can be zero, while the average Fisher information over different realizations is nonzero. This actually shows why the current problem should be put in the multivariate framework. When there is a random and

unknown phase ϕ in each realization, one actually needs to estimate two parameters g and ϕ . But it is easy to verify that they are not independent in this example, so the determinant of the Fisher information matrix is zero, hence the deviation of the estimator for both parameters is infinite, or in another word the "effective" Fisher information for both parameters is zero. This manifests the necessity of working in the multivariate framework for this problem.

Based on the above comments, this paper does not meet the criteria of Nature Communications. It might be suitable for other more specific journals with revision to make the result more solid.

Reviewer #3:

Remarks to the Author:

Authors have satisfactorily replied to the comments. In more detail, regarding the two points raised by Referee 2 in her/his last report:

1. I agree with Referee 2 that in general this estimation problem has to be considered as a multi-parameter one. However I believe that it is legitimate that the main focus on the manuscript has to be the estimation of the frequency difference ω_R , considering the other parameters as known. In fact, contrarily to other multi-parameter estimation problems that have been addressed in the literature, the main issue that one wants to address is the fact that, with standard estimation schemes, the error of the estimation of this parameter diverges, once the parameter goes to zero. If these standard estimation schemes are however working well for the other parameters characterizing the quantum statistical model (in this case ω_s and the noise parameter σ), I think that it is legitimate to consider a multi-step estimation protocol, where these parameters are previously estimated, and their value is then used to estimate the "problematic" parameter ω_r via the method suggested by the authors.

Of course I agree that a detailed and thorough study of the multi-parameter case could be useful and relevant both for theorists and experimentalists (and I strongly suggest the authors to elaborate in future works from what has been already described in the Supplemental Material), but in my opinion it goes beyond the scope of this article (whose primary merit is to put on firm mathematical grounds super-resolution quantum enhanced schemes, and give some relevant and practical examples), and it is not necessary to deserve publication in Nat. Comm.

2. As regards the second comment, I strongly believe that the authors are correct in considering the Fisher information corresponding to the averaged quantum state, and not the Fisher information, averaged over all the different random states.

If one follows Referee 2 advice, one would consider a completely different estimation problem, where for each measurement, the observer knows also the values of the parameters A_i and B_i , and can use them in the post-processing of their data. This information is in fact not known, and the only extra information known, that one can use in building the classical estimator from the measurement data, correspond to the noise parameter σ , characterizing the probability distribution of the random parameters A_i and B_i . As a consequence, the precision corresponding to any unbiased estimator in this setting, obeys the Cramer-Rao bound given by the Fisher information calculated via the averaged state (and thus characterized by the noise parameter σ).

A similar illuminating example can be found if we consider a standard noisy parameter estimation problem, such as frequency estimation of N qubit under Markovian dephasing, with an initial GHZ state:

it is known that the quantum Fisher information in the noiseless case will scale as N^2 , according to the Heisenberg limit. On the other hand if we calculate the QFI considering the noisy Markovian master equation, it will lose the Heisenberg scaling and grow as " N ", following the so-called standard

quantum limit.

However the above mentioned noisy quantum state, can be considered as an average state over all the possible stochastic trajectories, given a certain unravelling of the master equation. For example, if one considers the unravelling corresponding to continuous photo-detection, at each point of the time-evolution of each trajectory, one will still have a "GHZ-like" state with a random (but known!) relative phase between the two vectors of the pure state that depends on the number of "quantum jumps" observed during the evolution (see for more details: Albarelli et al., Quantum 2, 110 (2018)). Clearly in this case the QFI for each realization of the noisy state, i.e. for each trajectory, will be equal to the noiseless one, and one will get back the Heisenberg scaling.

For all these reasons and also given the relevance of the results presented in the manuscript, I am happy to recommend it for publication in Nat. Comm.

However I strongly suggest the authors to address the minor remarks I will outline in the following.

A. Following one of the referees' suggestion, the authors have generalized the condition on the exponent of ω_r , for one of the eigenvalues, in order to achieve super-resolution, writing the the inequality $1 < k \leq 2$.

However it looks to me that this has been well specified in the "Claim" in the main text, while both the proof and the supplemental material have not been updated. I suggest the authors to revise these parts.

B. The main results obtained by the authors consider the limit where ω_r goes to zero. In this limit the quantum state will change its rank, as one of the eigenvalue will go to zero.

Recently, there have been a couple of papers that discuss the possible discontinuity of the quantum Fisher information whenever the quantum state changes its rank by varying the value of the parameter to be estimated, see in particular

D. Safranek, Phys. Rev. A 95, 052320 (2017).

L. Seveso, et al., arXiv:1906.06185 [quant-ph]

In particular in this second paper, it is shown that in this limit both the classical and the quantum Cramér-Rao bounds may not hold.

While this is not a major issue for the problem considered by the authors (they consider this limit, but they are always interested in the estimation of some small but finite difference ω_R), I suggest them to discuss this possible problem, that may rise whenever ω_R is really equal to zero, that is whenever there is in fact only a single frequency characterizing the signal.

C. I have found the notation "n" used in Eq. (12) to identify the noise contribution, a little unfortunate, as the same letter has been widely used before as an integer parameter in the optimal estimation protocol. I suggest the authors to change this notation.

D. The authors correctly specify that the results on optical super-resolution imaging, already presented in the literature, are in fact a special case of the phenomenon discussed in the manuscript. It is my opinion (but I would leave the final decision about this to authors and editor) that it would definitely help the reader some extra discussion about this, describing in more detail how and why this is the case, at least in the Supplemental Material.

Reply - Overcoming resolution limit with quantum sensing

Referee 2

Referee: 1. I noticed that the authors made a multivariate analysis for ω_s and ω_r in the supplemental material. But a simple histogram is not a sufficient analysis. The core is how to most efficiently extract the information of the two parameters at the same time. As I said in the last report, the authors mainly focused on the estimation of ω_r , but why can ω_s be neglected after the initial ?

I understand that the purpose of this work is to enhance the distinguishability of two frequencies, but the analysis should be put on the framework of multivariate estimation, otherwise it would be senseless. In multivariate parameter estimation, the precision (or Fisher information) of different parameters are often complementary (Nat. Commun. 5, 3532 (2014) showed a very good example for phase and phase diffusion), so in the current problem, the precision of ω_s may influence the precision of ω_r . How the precision of these two parameters interplay with each other is critical to the simultaneous estimation of these parameters, and needs to be investigated in detail.

In fact, when the authors did a rough fit for the parameter ω_s initially, they indeed did it for all unknown parameters ω_s , ω_r and sigma. So why can the authors change to focus on ω_r alone after the initial estimation? The authors gave a reason in the reply as well as in the main text that the QFI is regular. This is far from sufficient. There are two issues here: (i) The regularity of QFI does not mean the other parameters do not have influence on ω_r , it just means that it is possible to estimate all parameters; how to optimally estimate these parameters is still unknown. (ii) The QFI is not always saturable. There is a very strict condition for the saturability of QFI. A detailed analysis about this problem can be found in Phys. Rev. A 94, 052108 (2016). In addition, I also want to refer to the original superresolution paper Phys. Rev. X 6, 031033 (2016) which treats this problem in an exact multivariate way.

2. For the problem of average over realizations of A_i and B_i , even there is only one measurement for each realization, the Fisher information should still be averaged over all possible realizations. The reason is whatever A_i and B_i occurs in reality for each measurement, they are always random, and the measurement results are also random, so the deviation of the estimator is random as well and needs to be averaged over the realizations and the measurement results. The average over measurement results will give Fisher information, and the average over realizations will give the average Fisher information which I mentioned in the last report. Averaging over the qubit state does not give any physical things, because this is not the state of the qubit in reality for any realization.

The authors gave an example that the average of a single realization with a random phase can be zero, while the average Fisher information over different realizations is nonzero. This actually shows why the current problem should be put in the multivariate framework. When there is a random and unknown phase ϕ in each realization, one actually needs to estimate two parameters g and ϕ . But it is easy to verify that they are not independent in this example, so the determinant of the Fisher information matrix is zero, hence the deviation of the estimator for both parameters is infinite, or in another word the "effective" Fisher information for both parameters is zero. This manifests the necessity of working in the multivariate framework for this problem.

Reply: We respectfully disagree, see comments of the third referee (and our reply). We extended the multivariate analysis, and added a more elaborate analysis of the FI matrix that matches the numerical estimation results.

Referee 3

Referee: I agree with Referee 2 that in general this estimation problem has to be considered as a multi-parameter one. However I believe that it is legitimate that the main focus on the manuscript has to be the estimation of the frequency difference ω_r , considering the other parameters as known. In fact, contrarily to other multi-parameter estimation problems that have been addressed in the literature, the main issue that one wants to address is the fact that, with standard estimation schemes, the error of the estimation of this parameter diverges, once the parameter goes to zero. If these standard estimation schemes are however working well for the other parameters characterizing the quantum statistical model (in this case ω_s and the noise parameter σ), I think that it is legitimate to consider a multi-step estimation protocol, where these parameters are previously estimated, and their value is then used to estimate the problematic parameter ω_r via the method suggested by the authors. Of course I agree that a detailed and thorough study of the multi-parameter case could be useful and relevant both for theorists and experimentalists (and I strongly suggest the authors to elaborate in future works from what has been already described in the Supplemental Material), but in my opinion it goes beyond the scope of this article (whose primary merit is to put on firm mathematical grounds super-resolution quantum enhanced schemes, and give some relevant and practical examples), and it is not necessary to deserve

publication in Nat. Comm.

Reply: We thank the referee for his comment. We extended the analysis of the multivariate part in the supplemental material (supplementary note 7). We explain that one needs to perform at least three different measurements (one for every parameter), and show that the FI matrix converges to a block diagonal matrix (as $\omega_r \rightarrow 0$). Since the full expression of the FI matrix is cumbersome, we didn't write explicitly all the matrix elements. Since it is block diagonal, we get that $\Delta\omega_r = \frac{1}{\sqrt{I_r}}$, where I_r is the same FI as in the main text. Namely if we repeat these 3 measurements many times we lose a factor of $\sqrt{3}$ because of the extra two measurements. These analytical results were verified with numerical estimation.

Referee: Following one of the referees suggestion, the authors have generalized the condition on the exponent of ω_r , for one of the eigenvalues, in order to achieve super-resolution, writing the the inequality $1 < k \leq 2$. However it looks to me that this has been well specified in the Claim in the main text, while both the proof and the supplemental material have not been updated. I suggest the authors to revise these parts.

Reply: We thank the referee for this comment, we changed the relevant parts accordingly.

Referee: The main results obtained by the authors consider the limit where ω_r goes to zero. In this limit the quantum state will change its rank, as one of the eigenvalue will go to zero. Recently, there have been a couple of papers that discuss the possible discontinuity of the quantum Fisher information whenever the quantum state changes its rank by varying the value of the parameter to be estimated, see in particular

D. Safranek, Phys. Rev. A 95, 052320 (2017). L. Seveso, et al., arXiv:1906.06185 [quant-ph]

In particular in this second paper, it is shown that in this limit both the classical and the quantum Cramr-Rao bounds may not hold. While this is not a major issue for the problem considered by the authors (they consider this limit, but they are always interested in the estimation of some small but finite difference ω_r), I suggest them to discuss this possible problem, that may rise whenever ω_r is really equal to zero, that is whenever there is in fact only a single frequency characterizing the signal.

Reply: We are grateful to the referee for this comment.

The issue the referee raises is: Suppose $p_i(\theta_0) = 0$ yet it doesn't vanish in an environment of θ_0 , then formally when calculating the FI for $\theta = \theta_0$ the terms that correspond to p_i should not be included (because it vanishes), and this creates a discontinuity. It should create also in our case, because these are exactly the non-zero terms. But then the second reference shows that this formal FI for $\theta = \theta_0$ is meaningless because the Cramer-Rao bound is violated in $\theta = \theta_0$: if we just estimate θ according to the estimation of p_i we get an unbiased estimator with zero variance (which should correspond to an infinite FI), which is in fact very good.

As the referee mentions, we are interested in the behavior of the variance/FI for small ω_r and not $\omega_r = 0$, therefore we are interested in the limit. However this means that we need to be more careful in the formulation the superresolution condition. The original formulation was $\frac{\partial p}{\partial \omega_r} = 0$ & $I_r > 0$, while a more careful and accurate one is $\frac{d p}{d \omega_r} \rightarrow 0$ and $I_r(\omega_r \rightarrow 0) > 0$. Therefore we changed the formulation accordingly. We added a comment (and citations) to the main text:

“An accurate formulation of the superresolution condition is $\frac{d p}{d \omega_r} \rightarrow 0$ and $I_r(\omega_r \rightarrow 0) > 0$, namely the limit needs to be positive. That is because we are interested in the behavior of the FI for a very small difference, rather than a vanishing difference. We mention this point since the FI at $\omega_r = 0$ can be discontinuous or meaningless (Cramer-Rao bound may be violated), as one of the eigenvalues vanishes [citations]. Given a vanishing eigenvalue, the variance of maximum likelihood estimation will vanish (which corresponds to an infinite FI) and thus may not coincide with the limit.”

Referee: I have found the notation n used in Eq. (12) to identify the noise contribution, a little unfortunate, as the same letter has been widely used before as an integer parameter in the optimal estimation protocol. I suggest the authors to change this notation.

Reply: We thank the referee for this comment, we changed the noise term to ε (in the main text and fig. 5).

Referee: The authors correctly specify that the results on optical super-resolution imaging, already presented in the literature, are in fact a special case of the phenomenon discussed in the manuscript. It is my opinion (but I would leave the final decision about this to authors and editor) that it would definitely help the reader some extra discussion about this, describing in more detail how and why this is the case, at least in the Supplemental Material.

Reply: The referee is right. We didn't want to write the details of the imaging problem because we didn't want to divert the focus from spectroscopy/general formulation to imaging (and we have a length limitation...). We added a section to the supplemental

material: “Relation to quantum superresolution in imaging” (supplementary note 3) . In this section we summarize the imaging scheme and show that it is a special case of this quantum resolution phenomenon. We refer to this section in the main text.